Subject Area:
neuroscience

Keywords:
rod photoreceptors, phototransduction, phosphodiesterase PDE6, dimeric activation, response kinetics, dominant time constant

Author for correspondence:
Trevor D. Lamb
e-mail: trevor.lamb@anu.edu.au

# A quantitative account of mammalian rod phototransduction with PDE6 dimeric activation: responses to bright flashes

Trevor D. Lamb[1] and Timothy W. Kraft[2]

[1]Eccles Institute of Neuroscience, John Curtin School of Medical Research, The Australian National University, Canberra, ACT 2600, Australia
[2]Department of Optometry and Vision Science, University of Alabama at Birmingham, Birmingham, AL, USA

 TDL, 0000-0003-0299-6115

We develop an improved quantitative model of mammalian rod phototransduction, and we apply it to the prediction of responses to bright flashes of light. We take account of the recently characterized dimeric nature of PDE6 activation, where the configuration of primary importance has two transducin molecules bound. We simulate the stochastic nature of the activation and shut-off reactions to generate the predicted kinetics of the active molecular species on the disc membrane surfaces, and then we integrate the differential equations for the downstream cytoplasmic reactions to obtain the predicted electrical responses. The simulated responses recover the qualitative form of bright-flash response families recorded from mammalian rod photoreceptors. Furthermore, they provide an accurate description of the relationship between the time spent in saturation and flash intensity, predicting the transition between first and second 'dominant time constants' to occur at an intensity around 5000 isomerizations per flash, when the rate of transducin activation is taken to be 1250 transducins $s^{-1}$ per activated rhodopsin. This rate is consistent with estimates from light-scattering experiments, but is around fourfold higher than has typically been assumed in other studies. We conclude that our model and parameters provide a compelling description of rod photoreceptor bright-flash responses.

## 1. Introduction

Vertebrate phototransduction has been studied extensively since the 1970s, and a number of quantitative molecular models have been developed that have provided a good description of many features of the rod's electrical response to light (e.g. [1–8]). However, we contend that each of these models suffers important shortcomings, which we enumerate below. In light of these issues, we have developed a new quantitative model of vertebrate phototransduction, and we have investigated its applicability to the electrical responses of mammalian rods to bright flashes of light.

We suggest that the most serious shortcoming in previous quantitative descriptions of the vertebrate phototransduction cascade is that they have invariably overlooked the dimeric nature of the activation of the PDE6 by transducin, as originally reported decades ago [9], and recently re-examined [10,11]. In order to simplify quantitative analysis, Lamb & Pugh [2] proposed that the activation of the PDE6 might be approximated as occurring independently for the two sub-units, and in a number of respects that approach has indeed been adequate. However, our recent examination of the implications of dimeric activation [11] and our further analysis in this paper have shown that the 'independent activation' simplification can lead to errors in the predicted kinetics of both activation and shut-off of the light response.

A second significant shortcoming of previous models involves the magnitudes of the parameter values that need to be assumed. Most importantly, we contend

royalsocietypublishing.org/journal/rsob  Open Biol. 10: 190241

that in the past the rate of activation of transducin by a single activated rhodopsin (R*) has been greatly underestimated, and various other parameters have been overestimated, in order for the models to correctly emulate the experimentally observed amplification of the single-photon response. Rather than the previously accepted rate of transducin activation in mammalian rods of approximately 300–350 G* s$^{-1}$ per R*, we shall present evidence that a more accurate estimate of the rate is around 4× higher, at approximately 1250 G* s$^{-1}$ per R*.

A key piece of evidence supporting this higher rate of transducin activation relates to the 'transition intensity' at which the change from a lower to a higher dominant time constant is observed to occur, in measurements of the time that the response remains in saturation. Using the previous rate of 300–350 G* s$^{-1}$ per R*, the transition intensity is calculated to be around 12 000–20 000 R* per rod (depending on exactly what other assumptions are made; see Discussion), whereas the observed transition intensity for a mouse rod is 4000–5000 R* per rod. This substantial discrepancy represents a third shortcoming of previous modelling, which we will show is eliminated under our revised model and with our revised parameter values.

We think that a fourth shortcoming of the independent activation model is that it predicts the first dominant time constant, $\tau_{D1}$, to be equal to the tail recovery time constant, $\tau_{rec}$, measured at late times in the response (i.e. $\tau_{D1} = \tau_{rec}$). However, experiments show a genuine discrepancy between these two measured parameters, with $\tau_{D1}$ roughly 20–25% larger than $\tau_{rec}$ (see, for example, [12,13]). Remarkably, we will show that such a ratio, of $\tau_{D1} \approx 1.2\, \tau_{rec}$, is indeed predicted by our dimeric PDE6 activation model.

A fifth shortcoming of the independent activation model is that it fails to explain an additional delay of approximately 5 ms (compared with cone responses) that is observed in the rising phase of rod responses to moderate flashes [8,11], a feature that is accounted for in the new model.

We suggest that another shortcoming of almost all previous modelling relates to the assumed time-course of R* activity. When modelling bright-flash responses, it has generally been assumed that R* decays exponentially, but there is no experimental evidence for this contention. On the other hand, for simulation of single-photon responses, it has instead typically been assumed that R* activity declines in multiple small steps, with each additionally attached phosphate group. However, we have shown that that assumption leads to non-physiological properties for the simulated single-photon responses [14], including an implausible shape for the amplitude distribution histogram and the occurrence of individual events with kinetics deviating considerably from those that are observed experimentally. An alternative model, in which phosphorylation instead simply alters the stochastic rate of abrupt R* shut-off, provides a better description of the single-photon response properties [14], and we will adopt that description here.

Finally, we note that the late component that is observed between 3 s and 10 s after flashes delivering 10 000–50 000 R* per rod is not accounted for in previous modelling. Here we will see that, by invoking the occurrence of a low rate of 'aberrant' R* shut-off events (as reported by [6,15,16]), it is possible to generate responses bearing close similarity to those measured experimentally from mammalian rods.

By combining these ideas, we have developed an improved quantitative model of mammalian rod phototransduction, and we have applied it to the prediction of responses to bright flashes of light, as follows. We consider the case of abrupt shut-off of

**Figure 1.** Reactions of transducin and phosphodiesterase at the disc membrane. Activation steps are in red, and shut-off steps are in blue. Activated transducin, G*, is created (dotted arrow) when activated R* is present. Molecules of the phosphodiesterase PDE6 exists in one of three forms: as E, with no G* bound; as E*, with one G* bound; or as E**, with two G*s bound. G* binds with E to form E* at rate $r_1$, and binds with E* to form E** at rate $r_2$; expressions for these rates are given in the text. The three shut-off steps each involve hydrolysis of the terminal phosphate of a G*. The singly bound E* shuts off with rate constant $k_{E*}$; the doubly bound E** shuts off with rate constant $k_{E**}$; and the unbound G* shuts off with rate constant $k_{G*}$.

R* [14], and we take account of the dimeric nature of PDE6 activation and shut-off, where the doubly activated form E** is of primary importance [11]. In analysing the time-course of $E^{**}(t)$, we model the case of multiple photoisomerizations per disc surface in three different ways, in each of which we take account of the decay of free (i.e. unbound) activated transducin (G*). By then solving the downstream (cytoplasmic) reactions, we predict the electrical response over a wide range of saturating flash intensities, and we show that the predictions account well for the qualitative form of experimentally measured responses of wild-type mouse rods to bright flashes. From these predicted electrical responses, we can plot the time in saturation ($T_{sat}$) as a function of flash intensity ($\Phi$, in photoisomerizations per rod) semi-logarithmically [17]. As seen in experiments, the model predicts two intensity regimes exhibiting dominant time constants, $\tau_{D1}$ and $\tau_{D2}$, of around 250 ms and 750 ms, respectively, when a 10% criterion is used to define recovery. Importantly, we obtain the correct transition intensity between regimes, of $\Phi_{trans} \approx 5000$ R* per rod, when we use a transducin activation rate of $v_{G*} = 1250$ G* s$^{-1}$ per R*.

## 2. Model

We begin by describing the reactions at the disc surface that underlie response recovery, and thereafter we outline our approaches to simulating these reactions in the case of bright flashes. Our simulation methods can be divided into three categories: (i) a very time-consuming 'full' approach that simulates the 2D diffusional interactions of reacting molecules at the disc surface; (ii) a much faster 'mass-action' approach that relies on the attainment of spatial homogeneity on the disc membrane; and (iii) a variant of the second case that additionally takes account of the low probability of stochastic occurrence of 'aberrant' R* shut-off events on different disc surfaces. The three approaches are now outlined, and details of the methods are given in §5, along with several analytical solutions that we obtained, some of which provide checks on the simulations. Thereafter, for evaluation of the rod's electrical response, our methods closely followed those in other recent studies, and are described in §5.12.

### 2.1. Disc-based reactions of transducin and the phosphodiesterase

The disc-based reactions involving transducin and the PDE are shown schematically in figure 1. Activation reactions

are shown in red, with the dotted arrow denoting G* formation at a rate proportional to R* activity, and the two curved red arrows indicating binding to form E* and E** at rates $r_1$ and $r_2$. The shut-off reactions are shown in blue, and each involves hydrolysis of the terminal phosphate of a G*·GTP, that is either (i) singly bound as E*, or (ii) doubly bound as E**, or (iii) unbound as free G*; the corresponding three rate constants are $k_{E*}$, $k_{E**}$ and $k_{G*}$.

## 2.2. Approaches to simulation of the disc-based reactions (methods 1, 2, 3)

*Method 1: Stochastic simulation of 2D diffusional interactions.* Our first approach to calculating the bright-flash responses was to undertake 'full' simulations of the 2D diffusional interactions between the molecules on the disc membrane, using the same methods as in our recent analyses of single-photon responses [11]. We refer to this as method 1, and it is described further in §5.1. Unfortunately, though, this approach becomes excessively time-consuming for intense flashes, because of the combination of the need to simulate multiple cases with different numbers of R* per disc surface, the need to simulate out to long post-flash times, and the need to repeat the simulations many times in order to achieve a reliable average. On the laptop computer that was available, it took around 17 h to obtain a good average for the $E**(t)$ response in the case of a fixed number of 10 R*s per disc surface. And because of the Poisson nature of photon absorption on different discs, we required comparable simulations at numerous other numbers of R*s per surface, in order to be able to calculate the electrical response to a bright flash. Therefore, we developed and validated a much faster approach.

*Method 2: Approximation in the case of spatial homogeneity of disc reactants.* Within approximately 100 ms of the delivery of a bright flash, the punctate activity of R* molecules will have subsided (see trace for R* in figure 2a), and within a total of a few hundred milliseconds the distributions of reactant species (G*, E, E*, E**) on the disc membrane will have each become spatially homogeneous (see §5.3.1). After that time, we can simply model the interactions between the molecules using a mass-action approach that ignores spatial effects. In §5.3 we present the differential equations applicable in this case, and we derive expressions for the rates $r_1$ and $r_2$ in figure 1 above. Section 5.5 then gives details of this approach, which we refer to as method 2. This method was more than 2000-fold faster than method 1, primarily because it required no simulation of spatial interactions. In the Results section, we compare the predictions of methods 1 and 2, and show close agreement between them. As a further check, in §5.6 we derive an analytical solution for a lower limit to the time-course for the decay of free G*, and we show that the simulated G* traces for method 2 are just marginally above this theoretical lower limit.

*Method 3: Addition of stochastic occurrence of 'aberrant' R* shut-off events.* If even a small proportion of activated R* fail to inactivate normally (i.e. within 200 ms), and instead take seconds to shut off, then this will markedly affect the recovery predicted at late times for very intense flashes. In particular, our simulations show a slow tail in the predicted recovery, that bears a marked resemblance to experimentally recorded late slow tails. However, because of the relatively small number of aberrant R* events (even at quite high flash intensities), and the fact that

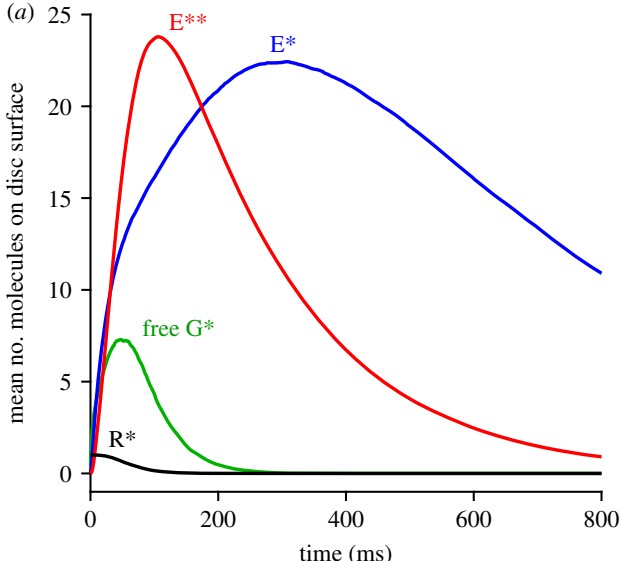

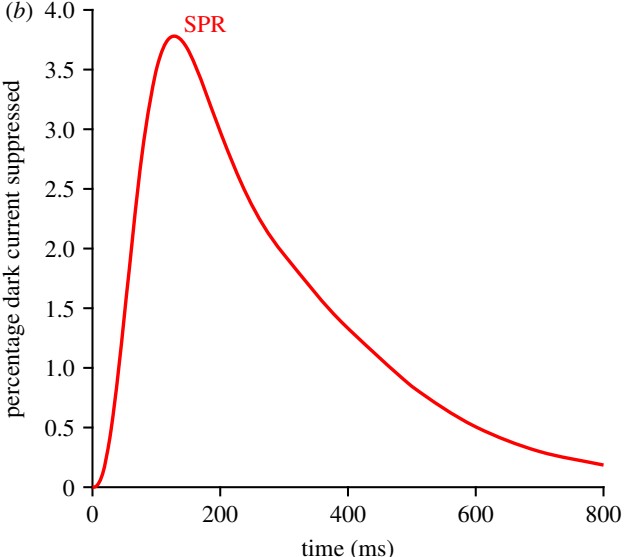

**Figure 2.** Predicted mean single-photon response (SPR). (*a*) Disc reactants. The full 2D simulation approach of method 1 was applied, using the standard parameters listed in table 1, and with a single R* created at a random position on the disc surface at time zero in each trial. Traces represent the mean responses to 2000 trials. (*b*) Electrical response. The downstream reactions were solved for the case of longitudinal diffusion in the outer segment, using the standard parameters listed in table 2, in response to each individual simulation of $E**(t)$ time-course used in deriving (*a*). The trace in (*b*) is the mean of those 2000 individual downstream responses.

these events occur at stochastic locations along the outer segment, it is unfortunately necessary to simulate the activity on each disc surface, and then subsequently integrate the downstream cytoplasmic equations for the spatial case, and as a result this method is slow. Section 5.10 gives further details of this approach, which we refer to as method 3.

## 2.3. Considerations relevant to the numerical simulations

### 2.3.1. Sources of fluctuation in the activation and shut-off steps

In approximating the response to a very bright flash, one might think it sufficient to analyse the downstream behaviour elicited by the mean level of activation across all the disc surfaces, but our analysis shows that this is not the case. A major

**Table 1.** Parameters for simulation of lateral diffusion reactions underlying dimeric activation of PDE**. The stimulus corresponded to a brief flash at $t = 0$ that delivered $Q$ photoisomerizations at random locations on the circular disc. G*s were generated stochastically using the 'shortcut' method (see text), at a mean rate $\nu_{G*}$ at the position of each R*; thus the density $C_G$ of G-protein holomers $G\alpha\beta\gamma$ was only needed in calculating depletion of G-protein. The initial numbers of G-protein and PDE holomers were the same in each trial (i.e. they were not stochastic variables). Reactions between diffusing molecules occurred at the simulated diffusion limit (i.e. upon each contact between molecules that could react with each other).

| symbol | description | value | units |
|---|---|---|---|
| | *dimensions and time increment* | | |
| $d$ | diameter of circular disc | 1.3 | µm |
| $\Delta x$ | lattice grid spacing | 5 | nm |
| $\Delta t$ | time increment | 0.5 | µs |
| | *lateral diffusion at the disc membrane* | | |
| $C_G$ | density of G-protein $G\alpha\beta\gamma$ on disc membrane | 2500 | µm$^{-2}$ |
| $K_m$ | Michaelis constant of G-protein depletion | 0.14 | |
| $C_E$ | density of PDE holomers on disc membrane | 80 | µm$^{-2}$ |
| $D_{R*}$ | lateral diffusion coefficient of R* | 1.5 | µm$^2$ s$^{-1}$ |
| $D_{G*}$ | lateral diffusion coefficient of G* | 2.2 | µm$^2$ s$^{-1}$ |
| $D_E$ | lateral diffusion coefficient of PDE | 1.2 | µm$^2$ s$^{-1}$ |
| $D_{E*}$ | lateral diffusion coefficient of PDE* | 1.0 | µm$^2$ s$^{-1}$ |
| | *stochastic R* shut-off* [14] | | |
| $M$ | minimum phosphates required before Arr binding | 3 | |
| $\mu$ | common rate of the $M + 1$ R* shut-off reactions | 60 | s$^{-1}$ |
| | *rates of transducin activation and PDE shut-off* | | |
| $\nu_{G*}$ | rate at which fully active R* creates G*s | 1250 | s$^{-1}$ |
| $k_{E*}$ | rate constant of PDE* decay to PDE | 2.5 | s$^{-1}$ |
| $k_{E**}$ | rate constant of PDE** decay to PDE* | 5 | s$^{-1}$ |
| $k_{G*}$ | rate constant of decay of unbound G* | 1.0 | s$^{-1}$ |

non-linearity is introduced in the system by the saturation of PDE activation that occurs at the very high levels of activated transducin elicited by bright flashes; this has the result that an alteration in the initial level $G*(0)$ of activated transducin does not cause a vertical scaling of $E**(t)$ activity, but instead causes an approximately horizontal shift. This effect is shown in the simulations of figure 6 in the Results section, where the number of photoisomerizations per disc surface differs in the three panels. Intuitively, one can see that when the PDE is 'overloaded' with excess transducin, then further overloading simply holds the PDE in saturation for even longer. A consequence of this non-linearity is that we need to simulate the activity occurring on individual disc surfaces as a result stochastic fluctuations, rather than simply taking the average across all discs.

One major source of fluctuation is stochastic variability in the number of photoisomerizations elicited on different disc surfaces, but another crucial source turns out to be stochastic variability in the lifetimes of individual activated rhodopsin molecules—and this variability needs to be considered even at very high flash intensities. Our subsequent analysis suggests that, by taking account of these two sources of fluctuation, we are able to obtain a good description of the rod's bright flash response. Other sources of variation that we do not take into account here are those downstream of R*, such as stochastic fluctuations in the rates of activation and shut-off of transducin and PDE6; these sources appear to be considerably less important because of the larger numbers of molecules participating.

### 2.3.2. Depletion of transducin $G\alpha\beta\gamma$ with very intense flashes

Another phenomenon that needs to be taken into consideration, though only at the very highest intensities, is depletion of the G-protein holomer $G\alpha\beta\gamma$. Intensities eliciting only a single photoisomerization per disc surface will elicit minimal depletion, even with the high rates of activation $\nu_{G*}$ analysed here. For a G-protein density of $C_G = 2500$ µm$^{-2}$ and an outer segment diameter of $d = 1.3$ µm, with a corresponding disc surface are of $A = 1.33$ µm$^2$, the total number of G-protein molecules per disc surface is around 3300. And for a mean R* lifetime of 68 ms (0.068 s) together with an activation rate of $\nu_{G*} = 1250$ G* s$^{-1}$ per R*, the mean number of G*s activated by the single R* would be 84, or just 2.5% of the complement of G-protein on that surface. However, for a very bright flash that delivered $Q = 40$ R* per disc surface, there would be near-total depletion of G-protein—at least, on the simplifying assumption that those different R*s acted independently of each other. This calculation makes it clear that, for modelling intense flash responses, we will need to determine the extent of depletion of transducin. This analysis is undertaken in §5.9.

### 2.3.3. Phototransduction cascade parameter values

The standard phototransduction cascade parameters that we settled upon, following preliminary analyses, are listed in table 1 for the disc-based reactions, and in table 2 for

royalsocietypublishing.org/journal/rsob    Open Biol. **10**: 190241

**Table 2.** Downstream phototransduction cascade parameters.

| symbol | description | value | units |
|---|---|---|---|
| | *parameters that determine the resting state* | | |
| $\beta_{Dark}$ | dark rate constant of cGMP hydrolysis | 4.0 | $s^{-1}$ |
| $\alpha_{max}$ | maximal rate of cGMP synthesis by GC | 150 | $\mu M\ s^{-1}$ |
| $f_{Ca}$ | fraction of CNGC current carried by $Ca^{2+}$ | 0.12 | |
| $K_{GCAP}$ | $Ca^{2+}$ concentration parameter of GCAP | 80 | nM |
| $m_{GCAP}$ | $Ca^{2+}$ cooperativity of GCAP | 1.5 | |
| $J_{cG,\ max}$ | maximal CNGC current for the OS | 2000 | pA |
| $n_{cG}$ | cooperativity of CNGC activation by cGMP | 3 | |
| $K_{cG}$ | cGMP concentration parameter of CNGC | 20 | $\mu M$ |
| $J_{ex,\ max}$ | maximal exchange current for the OS | 4.6 | pA |
| $K_{ex}$ | $Ca^{2+}$ concentration parameter of exchanger | 1100 | nM |
| | *calculated resting dark state* | | |
| $cG_{Dark}$ | dark cGMP concentration | 4.12 | $\mu M$ |
| $Ca_{Dark}$ | dark $Ca^{2+}$ concentration | 322 | nM |
| $\alpha_{Dark}$ | dark rate of cGMP synthesis by GC | 16.5 | $\mu M\ s^{-1}$ |
| $J_{Dark}$ | dark current | 18.4 | pA |
| | *parameters not affecting the resting state* | | |
| $\beta_{E**}$ | rate constant of cGMP hydrolysis by a PDE** | 0.017 | $s^{-1}$ |
| $L$ | length of outer segment | 22 | $\mu m$ |
| $N_{surfs}$ | number of disc surfaces per OS | 1320 | |
| $f_{cyto}$ | fraction of OS volume cytoplasmic | 0.5 | |
| $V_{cyto}$ | cytoplasmic volume of OS | 0.0146 | pL |
| $B_{Ca}$ | buffering power of cytoplasm for $Ca^{2+}$ | 50 | |
| | *longitudinal diffusion parameters (when used)* | | |
| $D_{cG}$ | longitudinal diffusion coefficient for cG | 40 | $\mu m^2\ s^{-1}$ |
| $D_{Ca}$ | longitudinal diffusion coefficient for $Ca^{2+}$ | 2 | $\mu m^2\ s^{-1}$ |
| $n_x$ | number of longitudinal elements simulated | 100 | |

reactions in the cytoplasm and at the plasma membrane. The great majority of these parameter values are unchanged from our recent study [11]. However, we raised the rate of transducin activation per R* to $v_{G*} = 1250\ G*\ s^{-1}$ (from our recently assumed value of $1000\ G*\ s^{-1}$), and lowered $\beta_{E**}$ to compensate (see below). We also adopted a slightly smaller diameter for the rod outer segment, of $d = 1.3\ \mu m$, based on recent cryo-anatomical studies [18,19]. In addition, we now need the number of disc surfaces per outer segment, which we take as $N_{surfs} = 1320$, calculated for an outer segment length of $L = 22\ \mu m$ using the number of discs per unit length of 30 discs $\mu m^{-1}$ determined in several recent studies [18–21].

# 3. Results

Prior to presenting our main results on the recovery of bright-flash responses, we begin by checking that the parameters specified in tables 1 and 2 provide a suitable description of the single-photon response, and then we examine the kinetics of the rising phase of the response.

## 3.1. Single-photon response

Figure 2*a* presents the mean simulated single-photon response at the level of the disc-based reactants (R*, G*, E* and E**) using the standard parameters in table 1. We standardized on a G-protein activation rate of $v_{G*} = 1250\ s^{-1}$ because (as set out in §3.6) this provided a good description of the transition intensity between dominant time constants in the bright-flash regime. This rate of activation is 25% higher than assumed in our recent analysis of single-photon responses [11] and, as shown by the red trace in figure 2*a*, it leads to a peak for E**(*t*) of 23.8 molecules in the mean single-photon response, which is about 32% larger than we obtained previously.

Therefore, in order to achieve a single-photon electrical response comparable with experiment, and with our earlier simulations, we reduced the hydrolytic activity for E** to $\beta_{E**} = 0.017\ s^{-1}$ (table 2), from our previous value of $\beta_{E**} = 0.024\ s^{-1}$. As shown in figure 2*b*, this generated a mean single-photon amplitude of 3.8% of the dark current, corresponding to 0.7 pA for a dark current of 18.4 pA, which is broadly consistent with experimental measurements

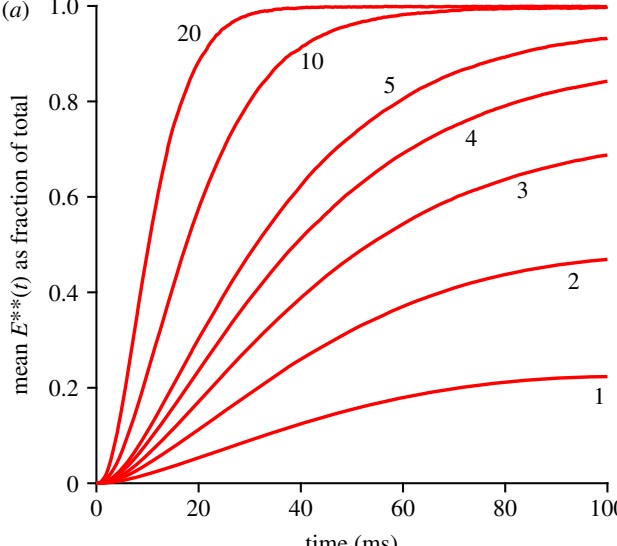

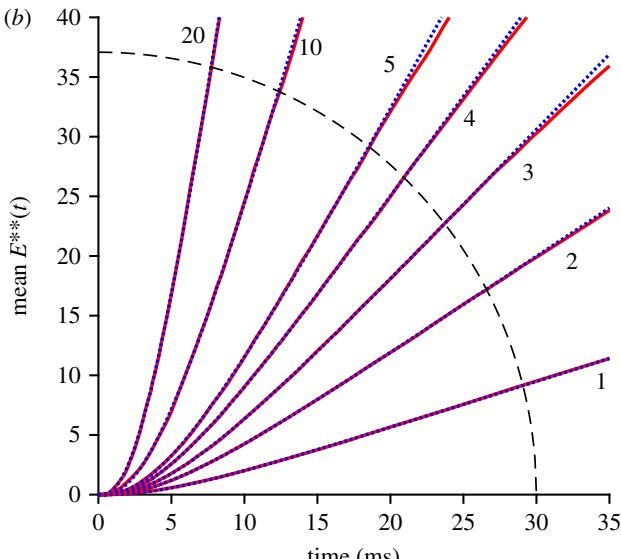

**Figure 3.** Mean rising phase for $E^{**}(t)$. (a) Fractional $E^{**}(t)$ as a function of time, for the indicated numbers of photoisomerizations per disc surface, $Q$. (b) Early rising phase of the same responses, fitted with the delayed ramp expression in equation (5.1). The magnitudes of the slope $\nu_{E^{**}}$ and the longer delay $\tau_2$ are plotted in figure 4; the shorter time constant was held constant at $\tau_1 = 0.8$ ms. The dashed arc shows the region outside of which data points were ignored in the fitting procedure.

from mammalian rods in the literature. Thus, exactly the same set of parameters that we use to describe mammalian rod bright-flash responses also provides a satisfactory description of mammalian rod single-photon responses.

### 3.2. Rising phase of bright-flash responses

We next investigated the early rising phase of PDE6 activity, $E^{**}(t)$, averaged from repeated runs with method 1, which simulates the 2D diffusional interactions using the WalkMat program. The collected mean $E^{**}(t)$ responses are shown in figure 3a for fixed numbers of photoisomerizations per disc surface, of $Q = 1, 2, 3, 4, 5, 10$ and $20$. As $Q$ increases, the mean $E^{**}(t)$ response becomes larger and rises earlier, while continuing to display an S-shaped onset. The form of that initial onset is examined more closely in the zoomed-in view of figure 3b, where the dotted blue traces are

least-squares fits of the theoretical function equation (5.1) that is derived in §5.2 (see below).

In previous work, for the case of a single photoisomerization ($Q = 1$), we fitted the onset phase of $E^{**}(t)$ with the expression for a ramp in time convolved with an exponential decay, representing a single delay stage of roughly 7 ms preceding the ramp-wise appearance of the doubly activated $E^{**}$; see eqn (2.1) of [11]. Here, though, with responses to multiple $R^*$s per disc surface (and with a 25% higher rate of activation), it became apparent that an additional shorter delay stage was also required, to provide an adequate description at very early times. Furthermore, we discovered that by fixing this shorter time constant at the expected first-contact time for activation of $G^*$ by $R^*$, namely at $\tau_1 = 1/\nu_{G^*}$, we could achieve a very good fit to the rising phase of the entire set of $E^{**}(t)$ responses. Thus, the fitted dotted blue traces in figure 3b plot the expression for a ramp in time (with slope $\nu_{E^{**}}$) convolved with two exponential decay stages having time constants $\tau_1$ and $\tau_2$, with the shorter of these ($\tau_1$) held constant.

We interpret the good fit of the 'doubly delayed ramp' to indicate that, at early times, the activation of PDE behaves as a linear process, that is delayed firstly by a very short time constant corresponding to the first contact time for collision of $R^*$ with a transducin, and delayed secondly by a somewhat longer time constant of roughly 5 ms. However, we do not have an intuitive explanation for why the dimeric activation of PDE to $E^{**}$ should behave in this manner. Instead, this is simply the observed behaviour when molecules of $G^*$ are sequentially activated locally at a high rate, and when two such $G^*$ molecules need to make diffusional contact with one molecule of PDE.

The fitted values that we obtained for $\nu_{E^{**}}$ and $\tau_2$ are plotted in figure 4. The upper panel shows that $\nu_{E^{**}}$ was essentially independent of $Q$, indicating that the initial rate of rise of $E^{**}$ remained directly proportional to the number of isomerizations per surface, even up to $Q = 20$. The lower panel shows that the time constant of the second delay, $\tau_2$, was roughly constant for $Q \leq 5$ $R^*$ per surface, corresponding to approximately 6000 $R^*$ per rod, but shortened somewhat at higher intensities. As a rough description of this shortening, we have drawn a Gaussian function, though we attach no significance to the actual shape used. In the Discussion we will consider the possible relevance of a shortening of this delay time constant at high intensities.

### 3.3. Recovery phase of PDE6 activity with multiple isomerizations per disc surface: comparison of methods 1 and 2

Our analysis of bright-flash recoveries in this section employs the relatively fast homogeneous approximation (method 2) to simulate the mean time-course of recovery of $E^{**}(t)$ for fixed numbers, $Q$, of isomerizations per disc surface. First, though, we check the predictions from that method against the very time-consuming 2D diffusion simulations (method 1), for a number of test values of $Q$. Then in the next section we consider the Poisson distribution of isomerizations received by different disc surfaces, in response to bright flashes uniformly illuminating the outer segment, to derive the mean $E^{**}(t)$ activity per outer segment, and then we integrate the downstream equations to predict the electrical response. Subsequently, in

royalsocietypublishing.org/journal/rsob    Open Biol. **10**: 190241

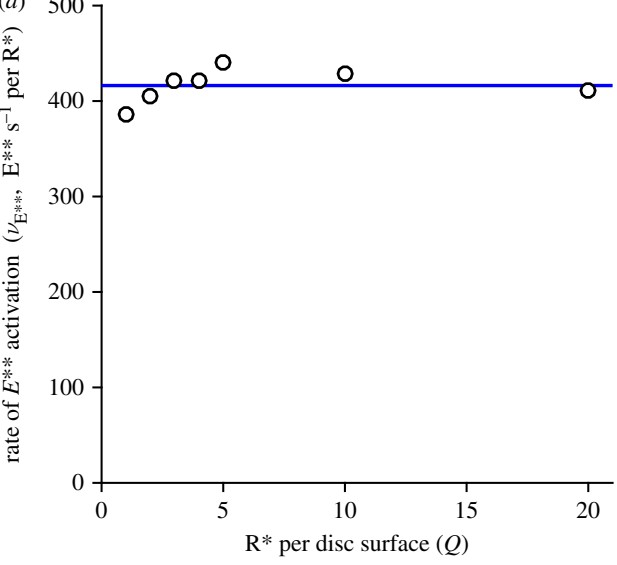

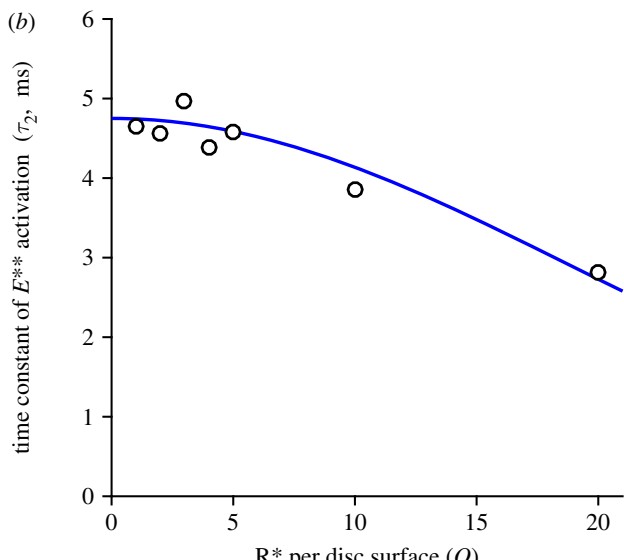

**Figure 4.** Fitted slope $\nu_{E**}$ and delay $\tau_2$ for the rising phase of $E**(t)$. The parameters obtained from the fitting of equation (5.1) in figure 3b are plotted against $Q$, the number of R*s per disc surface; the shorter time constant $\tau_1$ was held constant at 0.8 ms. The horizontal line in (a) is positioned vertically at the mean of the fitted values for $\nu_{E**}$, of 416 $E** s^{-1}$ per R* per surface. The curve in (b) is a Gaussian, though we attach no particular significance to the particular form used.

§3.5, we use quite a different approach (method 3) to simulate the bright-flash responses when a small proportion of isomerizations generate 'aberrant' R* shut-off events. Finally, in §3.6, we measure and plot the time $T_{sat}$ that each response remains in saturation, as a function of flash intensity Φ on a logarithmic scale, in order to investigate the dominant time-constant behaviour of rod phototransduction.

### 3.3.1. Sample recoveries for disc-based reactants using methods 1 and 2

Figure 5 illustrates our two approaches to calculating the recovery phase of the disc-based reactions elicited by multiple photoisomerizations per disc surface. Each panel presents a sample set of 25 stochastic simulations for a representative case of $Q = 10$ photoisomerizations per disc surface, with colour coding: free G*, green; E, black; E*, blue; and E**,

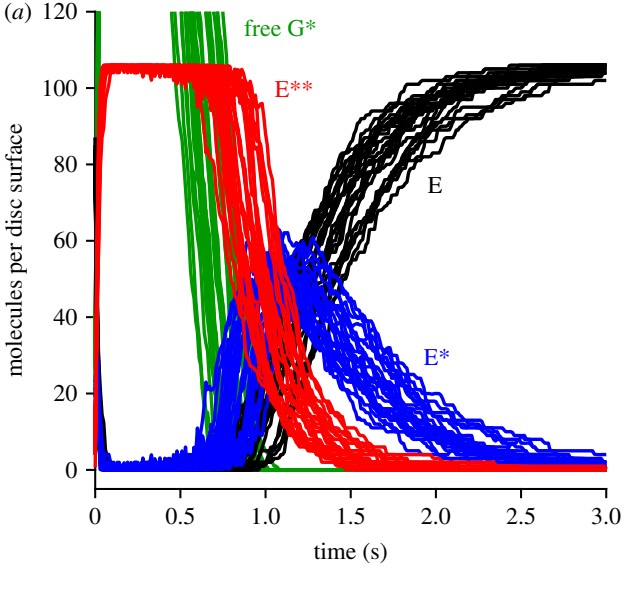

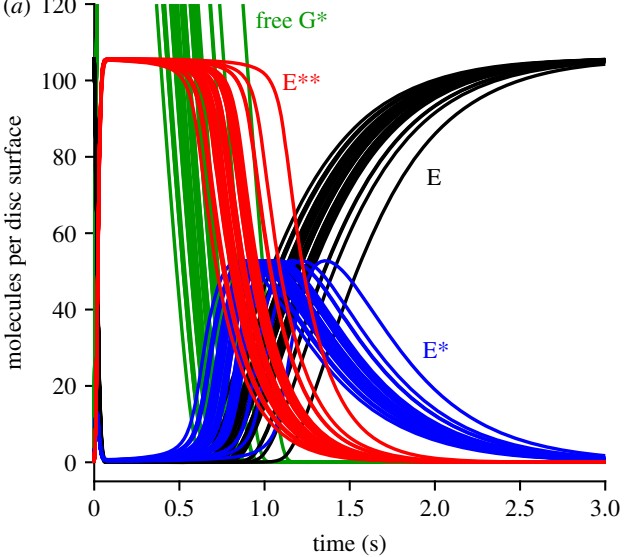

**Figure 5.** Sample simulations for the recovery phase of disc-based reactions for $Q = 10$ photoisomerizations per disc surface. (a) Full 2 D stochastic simulations using method 1. (b) Faster approach assuming spatial homogeneity using method 2. Each panel show the results of the first 25 simulations. All parameters were as listed in table 1. Note that the PDE is completely saturated when all 106 molecules per surface are in the E** state (as occurs with this initial 15-fold excess of free G*).

red. Figure 5a was obtained using the 2D spatial simulations of the WalkMat program (method 1), whereas figure 5b was obtained using the faster spatially homogeneous macroscopic approach described in §5.4 (method 2). The sets of traces in the two panels appear qualitatively similar, apart from the absence in the lower panel of stochastic fluctuations in the numbers of molecules, as would be expected for any macroscopic approach.

However, there was a huge difference in computation time, with each simulation taking approximately 5 min in the upper panel (method 1), compared with approximately 120 ms in the lower panel (method 2); this was on a 2.8 GHz laptop PC, running Matlab R2016a under Windows 10. Accordingly, a set of 200 simulations for $Q = 10$ with method 1 took around 17 h, and so with that method we restricted ourselves to just a handful of values of $Q$; in contrast, we could run 1000 simulations of the macroscopic approach in approximately 2 min using method 2.

royalsocietypublishing.org/journal/rsob   Open Biol. **10**: 190241

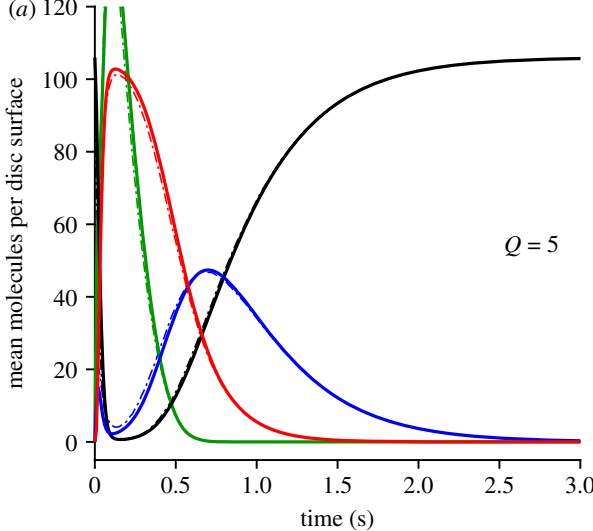

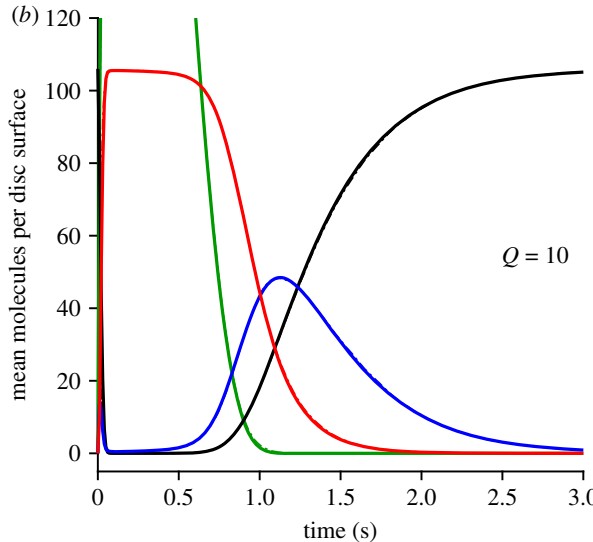

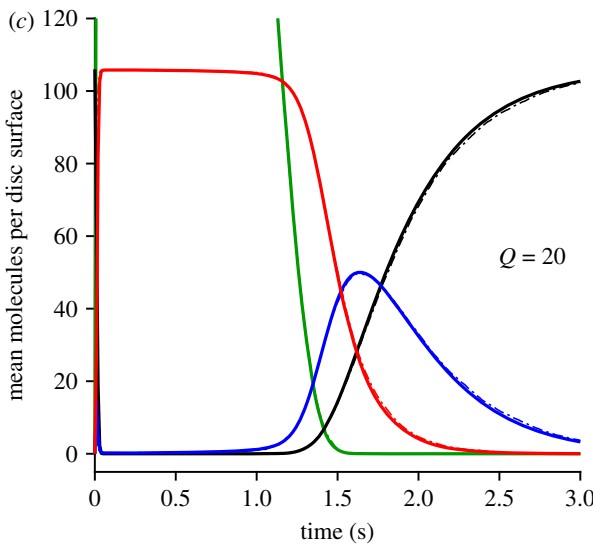

**Figure 6.** Mean levels of disc-based reactants $G^*$, $E$, $E^*$ and $E^{**}$ (a) $Q = 5$, (b) $Q = 10$ and (c) $Q = 20$ R* per disc surface. Continuous traces are for method 2, while the thinner dot-dash traces (that are mostly obscured) are for method 1. Green, free $G^*(t)$; black, $E(t)$; blue, $E^*(t)$; red, $E^{**}(t)$.

Inspection of the traces in figure 5 highlights the importance of taking account of stochastic fluctuations in R* lifetime in the macroscopic approximation. Thus, the differences between the individual traces in figure 5b stem from

the different levels of transducin activity elicited in the different runs, which in turn resulted from differences in the summed activity of the $Q$ individual R* molecules, even though every trial corresponded to the same number of isomerizations, $Q = 10$. Because of this variation in integrated R* activity, and the resultant temporal dispersion of individual recoveries, the mean $E^{**}(t)$ response (shown in figure 6b, red trace) has a shallower decline than that for the individual simulations. Had we instead assumed that the macroscopic response could have been calculated using the mean R* lifetime, then we would incorrectly have obtained a decline of $E^{**}(t)$ that was too steep.

### 3.3.2. Mean disc-based activity as a function of isomerizations per surface

The mean activity calculated over repeated simulations is plotted in figure 6, for three sample values of the number of isomerizations per disc surface ($Q = 5$, 10 and 20). The continuous traces are averages from method 2, whereas the thinner dot-dashed traces are averages from method 1, and in each of the three panels the two methods generate very similar mean responses, as demonstrated by the fact that the dot-dashed traces are mostly obscured. In fact, for $Q = 10$ and 20, we think it possible that the macroscopic approximation approach of method 2 may have generated marginally more accurate mean responses, because of the larger number of stochastic simulations that we conducted (1000) compared with the more modest numbers of simulations using method 1 (200 and 100 in the lower two panels). On the other hand, for $Q = 5$ the predicted $E^{**}(t)$ mean response did not quite reach saturation, with the result that recovery had begun before spatial homogeneity was likely to have been achieved, and so in this case (as well as for smaller values of $Q$) the assumptions underlying the macroscopic approximation in method 2 would not have been fulfilled. Accordingly, for these lower intensities, it is necessary to use method 1 rather than the approximate approach.

Comparison of the three panels in figure 6 shows that the main effect of increasing $Q$ is to shift recovery to later times. This shifting can also be seen in the collected traces for the mean $E^{**}(t)$ in figure 7a where, to avoid overcrowding, only a selection of values of $Q$ have been presented, as listed in the legend. The red traces are for method 2 and, although only selected traces are shown, we calculated the mean recovery kinetics for every value of $Q$ from 1 to 60. The black traces are for method 1 and, because of the time-consuming nature of this method, we only calculated responses for $Q = 1 \ldots 5$, 10 and 20. Even though the approximation of spatial homogeneity (upon which method 2 is based) is unlikely to be applicable for the first several hundred milliseconds, the red traces for $Q = 5$, 10 and 20 all provide reasonable descriptions of the corresponding black traces for method 1. And perhaps surprisingly, the red traces for $Q = 1$, 2, 3 and 4 are also quite good approximations to the black traces.

In considering the quality of fit of the red traces in figure 7a as a description of recovery of the rod's bright-flash response, we are in fact only interested in the final decline, below about 5 $E^{**}$ per disc surface, because it is only then that the rod's electrical response escapes saturation. Accordingly, we examined the tail phase of the predicted recoveries of $E^{**}(t)$ in the semi-logarithmic plot of figure 7b, for the same traces as in panel a. For each value of $Q$, the

royalsocietypublishing.org/journal/rsob    Open Biol. **10**: 190241

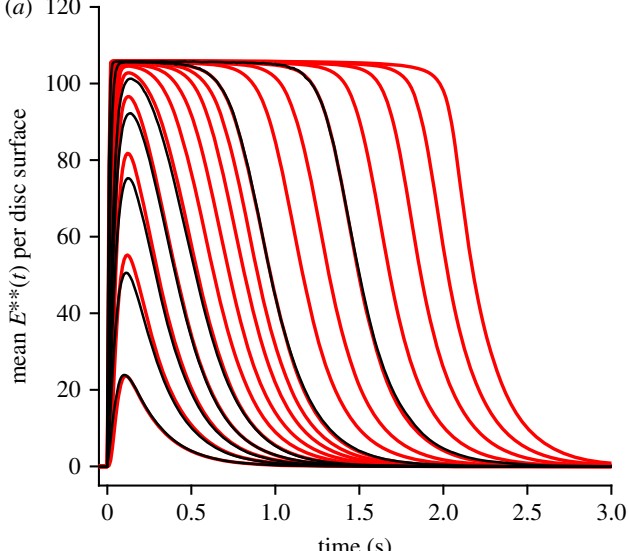

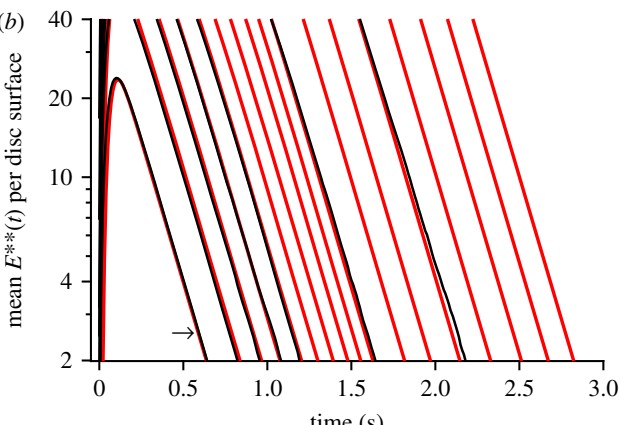

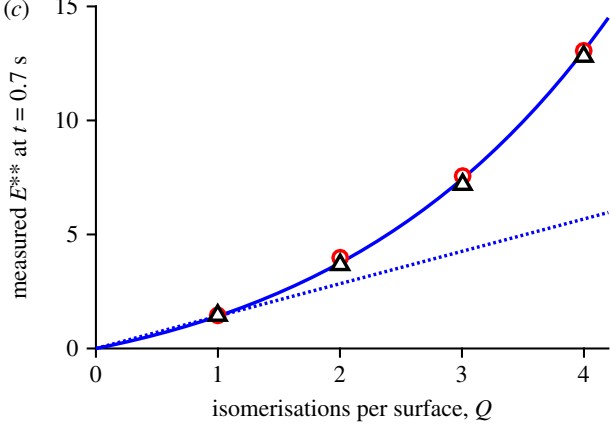

**Figure 7.** (a,b) Collected mean $E^{**}(t)$ activity for $Q$ up to 60 R* per disc surface, plotted (a) linearly and (b) semi-logarithmically. Black traces are for method 1, with $Q = 1 \ldots 5, 10, 20$. Red traces are for method 2, with $Q = 1 \ldots 10, 13, 16, 20, 25, 32, 40, 60$. All parameters are as listed in table 1. The slopes of the red traces in (b) each correspond to a time constant of 200 ms. For the arrow in (b), see text. (c) Response amplitudes measured at $t = 700$ ms; black triangles, method 1; red circles, method 2. Continuous blue trace plots the function $R_1 Q (1 + \rho)^{(Q-1)}$, with $\rho = 0.32$ and $R_1 = 1.42 E^{**}$; see equation (5.16). Dotted blue trace is a straight line, $R_1 Q$; i.e. for $\rho = 0$.

red traces for method 2 are very nearly straight lines, with a common slope corresponding to a rate constant of decline of $k_{E^{**}} = 5$ s$^{-1}$ (or a time constant of 200 ms). The seven black traces for method 1 are sufficiently similar to the red traces

to give us confidence that methods 1 and 2 predict closely similar recoveries in this small-signal region that is relevant to the post-saturation electrical response.

### 3.3.3. Super-linearity of the PDE6 response

A noteworthy feature of the $E^{**}(t)$ traces in figure 7 is super-linearity of response amplitude as a function of the number $Q$ of isomerizations per disc surface. This property is barely discernible in the linear plot of figure 7a, because the underlying super-linearity is counteracted by response compression as the $E^{**}(t)$ amplitude approaches the total number of available PDE6 molecules, $E_{tot} = 106$. The amplitudes of the tail responses (in panels $a$ or $b$), measured at a fixed time of 700 ms, are plotted in figure 7c as a function of $Q$, and it is clear that the values increase more steeply than the linear relation shown by the dotted line. The continuous blue trace plots the function $R_1 Q (1 + \rho)^{(Q-1)}$, with 'super-linearity parameter' $\rho = 0.32$, and with the single-quantum amplitude $R_1 = 1.42 E^{**}$ per surface. This relation, which is set out subsequently in equation (5.16), provides a good description of the measurements for $Q = 1, 2, 3$ and 4, at $t = 700$ ms. Because of the parallel nature of the tails, the same equation (with an appropriate value for $R_1$) also applies at other times, provided that the measurement time avoids the first few hundred milliseconds where method 2 is not valid, and provided that the amplitudes are not so large that response compression contributes appreciably.

This pronounced super-linearity of the response falling phase stands in contrast to the linearity that we found for the rising phase of the responses in figures 3b and 4a, and the difference can be appreciated intuitively in the following terms. For the rising phase, where the activation of transducin occurs in a punctate manner at each R* location, the level of activation is essentially independent for each R*, and hence the total activity at any instant is directly proportional to the number of photoisomerizations. But for the falling phase, where there has been spatial equilibration of the levels of proteins on the disc membrane, the intrinsic super-linearity of the PDE6's dimeric activation by two transducin molecules is exposed.

One consequence of this super-linearity, that is inherent in the dimeric model of PDE6 activation, turns out to be a significant deviation from the notion of a 'dominant time constant', as used in conventional analyses of the relationship between flash intensity and the time $T_{sat}$ that the electrical response spends in saturation [17]. This effect is examined analytically in §5.8, and the prediction obtained there is compared with simulation and with experiment in §3.6. The position of the arrow in figure 7b is used in that analysis to predict the vertical position of the theoretical curve.

### 3.4. Family of bright-flash responses (method 2)

To predict the entire time-course of the mean $E^{**}(t)$ response, summed over all the discs of the rod outer segment, we took the predicted responses for defined numbers of isomerizations per disc surface in figure 7 and weighted them according to the Poisson distribution, to generate the traces shown in figure 8a. These traces for mean $E^{**}(t)$ over the outer segment broadly resemble the previous traces for individual values of $Q$, though the recoveries are shallower because of the 'blurring' caused by averaging across traces for multiple values of $Q$.

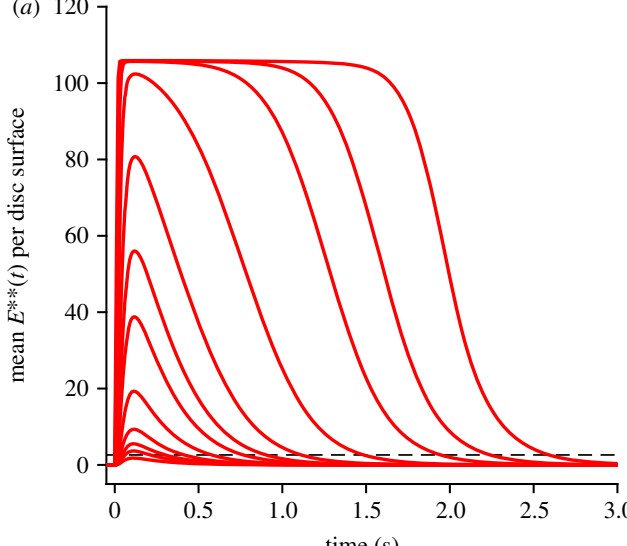

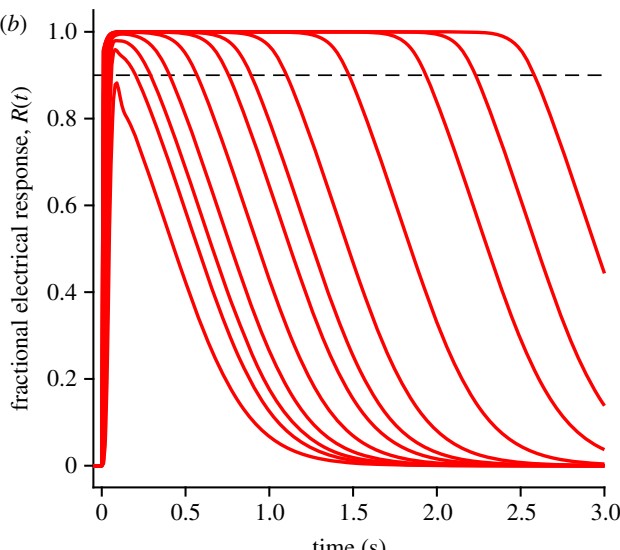

**Figure 8.** Mean E** activity over all the disc surfaces of the outer segment (*a*), and mean electrical response (*b*), calculated using method 2, for a series of bright flashes. Flash intensities were 100, 200, 300, 500, 1000, 2000, 3000, 5000, 10,000, 20,000, 30 000 and 50 000 R* per outer segment. All parameters were as listed in tables 1 and 2. Dashed horizontal lines show the 10% criterion level for recovery of the electrical response from saturation in (*b*), and the corresponding level of 2.6 E** per disc surface in (*a*).

Using the $E^{**}(t)$ traces in figure 8*a*, the equations for the downstream reactions were then integrated, to generate the rod's electrical response (figure 8*b*). The two panels in figure 8 use the same set of flash intensities, with Φ running from 100 R* to 50 000 R* per rod per flash.

The dashed horizontal line in figure 8*b* shows the level of 10% recovery of circulating current, that we will subsequently use to define the time in saturation, $T_{sat}$. The corresponding dashed horizontal line in figure 8*a* shows the mean level of PDE6 activity, approximately 2.6 E** per disc surface, at which the electrical response escapes saturation, and its position emphasizes that only the final tail of $E^{**}(t)$ is relevant to the recovery of the electrical response.

As a modification of the downstream integrations, we tried replacing the mean $E^{**}(t)$ traces obtained using method 2 with those obtained using method 1, for $Q = 1 \ldots 5$, because figure 7*a* showed minor differences between the two methods.

However, the only discernible difference in the calculated responses for the outer segment was the introduction of small fluctuations in the late recovery of the electrical response at intermediate intensities, which we attribute to fluctuations in the tail traces of $E^{**}(t)$ for method 1 in figure 7*b*. Therefore, we chose to use $E^{**}(t)$ traces only from method 2 in all the downstream integrations.

## 3.5. Late slow recovery: inclusion of aberrant R* shut-off events (method 3)

Up to this point, our simulations have not attempted to incorporate any description of the late slow phase of recovery that is seen with flashes exceeding a few thousand R*. But, as it is possible that the presence of this late component will delay the escape of the electrical response from saturation, we decided to attempt a description. We based this preliminary description on the occurrence of 'aberrant' R* shut-off events [6,15,16] reported in the electrical response to flashes delivering approximately 10–1000 isomerizations, together with the assumption that comparable activity likewise occurs at even higher intensities. Our simulation approach is described in §5.10, and is termed method 3.

Three additional parameters are needed to describe the rod's response to aberrant R* shut-off events: the probability $p_{aberr}$ that each R* will generate such an event; the mean lifetime $\tau_{aberr}$ of these stochastic shut-off events; and a measure of their plateau amplitude. In monkey rods, Kraft & Schnapf [16] reported $p_{aberr} \approx 1/400$ and $\tau_{aberr} \approx 6.5$ s, with a plateau level of approximately 0.8 pA for a rod with a maximal response of 25 pA. Perhaps surprisingly, this plateau level is similar to the peak amplitude of the normal single-photon response, rather than being around twice its amplitude, as might be expected if the R* were to retain full activity during its aberrant lifetime; later we will consider the significance of this value. In our simulations, the way in which we set the plateau amplitude was by specifying the fractional activity $a_{aberr}$ of the R* during its aberrant lifetime, with $0 < a_{aberr} \leq 1$; in order to achieve a plateau level approximately equal to the peak of the single-photon response, we set $a_{aberr} = 0.25$. For the other two parameters, we set $p_{aberr} = 0.002$ and $\tau_{aberr} = 4$ s, similar to the values above for monkey rods, because we found that this choice provided a reasonable qualitative description of the late tail phase in experiments from the literature (see electronic supplementary material, figure S1, panels E and F).

The predictions of method 3, with the inclusion of aberrant R* events occurring at low probability, are shown by the red traces in figure 9, for the same set of flash intensities as in figure 8 but on an extended time-base; for comparison the blue traces show the predictions without aberrant R* events, using method 2 (and are therefore the same as in figure 8). For the mean $E^{**}(t)$ traces in the upper panel, it might be thought that the aberrant events appear to contribute little, yet for the mean electrical responses (red traces in the lower panel) it is clear that the aberrant events have a substantial impact at late times for intense flashes. Furthermore, these predicted traces bear a strong qualitative resemblance to intense-flash responses reported in a number of studies in the literature (see electronic supplementary material, figure S1), as will be considered in the Discussion.

The mean responses with the inclusion of aberrant events (red traces in figure 9) are each averaged from at

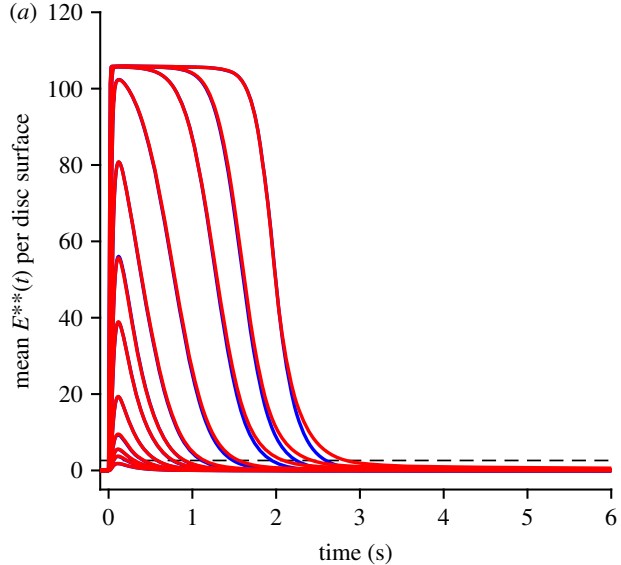

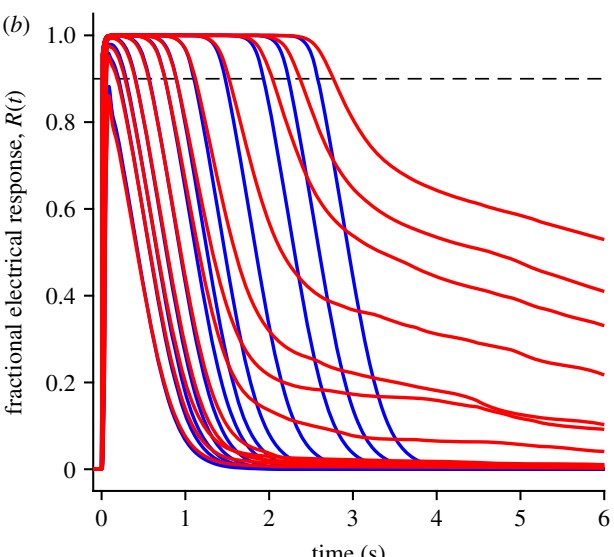

**Figure 9.** Inclusion of aberrant R* shut-off events, on slower time-base, for the same flash intensities as in figure 8. (*a*) Mean E** activity. (*b*) Electrical response. Blue traces are in the absence of aberrant events, and plot the same responses as in figure 8. Red traces are from method 3, with the inclusion of aberrant R* shut-off events occurring with a probability of 0.002, a mean lifetime of 4 s, and with activity 25% that of fully activated R*. The number of trials averaged was always at least 10, but was increased to 10 000/Φ for Φ < 1000; e.g. 50 trials averaged for Φ = 200 R*.

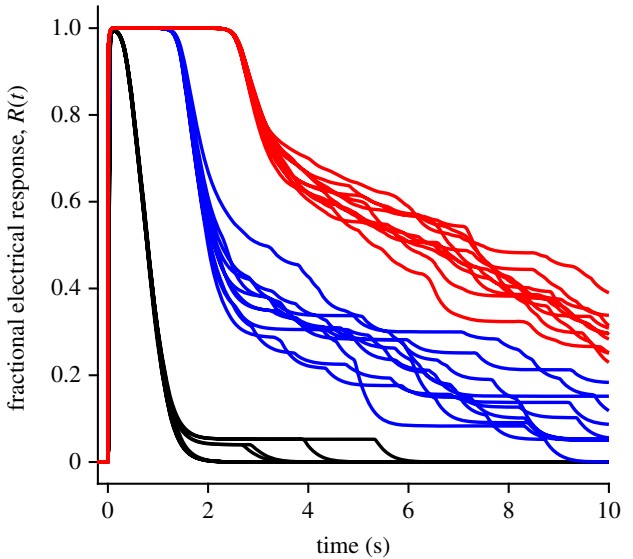

**Figure 10.** Fluctuations in raw responses to bright flashes, when aberrant R* events are included. For each of three intensities, the first 10 simulated responses are shown. Black, 500; blue, 10 000; red, 50 000 R* per outer segment. For 500 R*, six of the trials elicited no aberrant event, and superimpose.

## 3.6. Dominant time constant of recovery

For the bright-flash response recoveries predicted by methods 2 and 3, we measured the time $T_{sat}$ taken for the mean electrical response to escape saturation, defined here as recovery to 10% of the circulating current (i.e. 90% remaining suppressed). We made these measurements not only for the traces shown in figure 9*b*, but also for additional traces that are not illustrated, obtained at more closely spaced intensities. Our measured values of $T_{sat}$ from the simulations are plotted against flash intensity semi-logarithmically in figure 11, using blue for method 2 (which does not include any late slow component of recovery) and red for method 3 (which includes aberrant R* shut-off events). For intensities up to 5000 R*, the blue and red curves virtually superimpose, demonstrating internal consistency between the methods.

The heavy dotted black trace (shown only up to 10 000 R*) is the prediction obtained from equation (5.18) with $\rho = 0.32$, based on the observed super-linearity in the scaling of the final tails of E**(t) recovery in figure 7 (see §5.8). The thinner dotted straight lines have been fitted by eye, and have slopes corresponding to time constants of 245 and 780 ms; these lines intersect at $\Phi \approx 4900$ R*. Finally the symbols are the measurements plotted by Burns & Pugh [22].

The heavy dotted black curve, predicted from super-linearity of the tails of E** recovery using equation (5.18), provides an excellent qualitative description of the intensity dependence of $T_{sat}$ up to at least $\Phi \approx 3000$ R* per flash. Furthermore, both the blue curve and the red curve, obtained for simulations without and with aberrant R* events, provide a reasonable description of the experimental data across the entire range of flash intensities investigated, up to approximately 50 000 R* per flash. Furthermore, the blue and red curves both show a transition from a shallower to a steeper slope, and they both show some degree of curvature over the entire intensity range. Thus, for the parameter values that we have chosen, our simulations of the dimeric model of PDE6 activation provide a perfectly acceptable description of the experimentally observed dependence of $T_{sat}$ on flash intensity for mouse rods.

least 10 simulation trials, and they therefore obscure the magnitude of the underlying fluctuations. To illustrate these fluctuations, figure 10 plots a sample of 10 raw simulated responses for three representative intensities: 500, 10 000 and 50 000 R*. At the lowest of these intensities (black traces), individual aberrant R* shut-off events are clearly visible, whereas at the higher intensities (blue, red) the events overlap, so that one sees fluctuations rather than clear-cut transitions. The plateau level for the lowest intensity, of 4–5% of the dark current, is a direct result of our choice of an R* activity level of $a_{aberr} = 0.25$ in the aberrant state. However, the magnitude of this parameter is not crucial, because we found the simulated responses to be qualitatively similar for different values of $a_{aberr}$, though of course the plateau level and the amplitudes of the fluctuations were larger when $a_{aberr}$ was increased.

royalsocietypublishing.org/journal/rsob Open Biol. 10: 190241

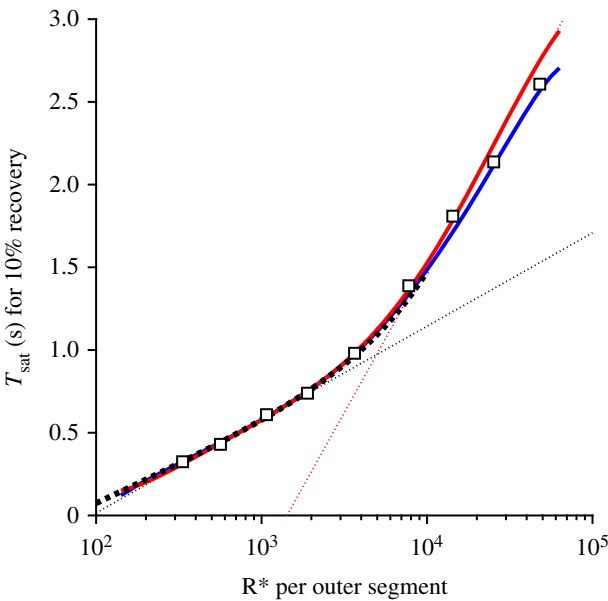

**Figure 11.** Saturation time $T_{sat}$ versus flash intensity $\Phi$, semi-logarithmically. Symbols are experimental measurements for mouse rods from fig. 4B of Burns & Pugh [22]. The blue and red curves are for method 2 (without aberrants) and method 3 (with aberrants), respectively, with parameter values as listed in tables 1 and 2. For both methods, the flash intensities ranged from 100 to 62 500 R*. For method 2 the intensities were spaced at intervals of 0.05 $\log_{10}$ units. For method 3 they were spaced at 0.1 $\log_{10}$ units and the smooth red curve is a fifth-order polynomial fitted through the measurements. For the aberrant R* events, the probability of occurrence was $p_{aberr} = 0.002$, the mean stochastic lifetime was $\tau_{aberr} = 4$ s and the activity relative to full R* activity was $a_{aberr} = 0.25$. The heavy dotted black curve plots equation (5.18) with $\rho = 0.32$ and vertical shift 585 ms. The dotted straight lines have slopes corresponding to dominant time constants of $\tau_{D1} = 245$ ms and $\tau_{D2} = 780$ ms, and intersect at a transition intensity of $\Phi_{trans} \approx 4900$ R* per rod.

At intensities around $\Phi = 1000$ R*, the tangent to the simulated curves has a slope of approximately 245 ms (see dotted line), in contrast to the time constant of 200 ms that we used for the decay of E** ($\tau_{E**} = 1/k_{E**}$, where $k_{E**} = 5$ s$^{-1}$). By way of comparison, we measured the tail phase of the simulated electrical responses, and found that these all had a final decay time constant of approximately 208 ms, marginally longer than $\tau_{E**}$. As a check, we also measured the tails of the $E**(t)$ traces (figure 7b), and confirmed that these all had a time constant of 200 ms, as expected. These measurements show that the dimeric model of PDE6 activation predicts an apparent first dominant time constant, $\tau_{D1}$, that is around 20% greater than the time constant of the tail of the electrical response, $\tau_{rec}$. In the Discussion we consider the significance of this difference.

# 4. Discussion

We consider the most important outcome of this study to be our demonstration that, with appropriate choice of parameter values, the recently proposed model of dimeric activation of the rod phosphodiesterase PDE6 [10,11] is able to provide a compelling description of bright-flash responses recorded in experiments on mammalian rods. Not only is there a close qualitative similarity to the form and kinetics of families of flash responses reported in many studies in the literature

(see §4.2 below), but in addition the relationship between flash intensity and time spent in saturation is accounted for accurately. Furthermore, the super-linearity that is intrinsic to the model predicts that the first dominant time constant of recovery, $\tau_{D1}$, should be around 20–25% greater than the time constant of recovery of the tail of the electrical response, $\tau_{rec}$, as is observed experimentally, with $\tau_{D1} \approx 240$ ms and $\tau_{rec} \approx 200$ ms. In addition, the simulated responses account for the occurrence of an additional delay in the rising phase. Achievement of the illustrated level of agreement between simulation and experiment requires constraints on the values of certain parameters, including the rate of transducin activation per R* and the number PDE6 holomers per disc surface, as we discuss below in §4.3.

## 4.1. Initial delay in the rising phase of the rod's flash response

In our previous study, we examined the rising phase of the simulated response to a single photoisomerization ($Q = 1$) for a rate of transducin activation of $v_{G*} = 1000$ G* s$^{-1}$, and we reported a delay corresponding to a time constant of approximately 7 ms. Here we have extended that analysis to the case of multiple photoisomerizations per disc surface (up to $Q = 20$), using a higher activation rate of $v_{G*} = 1250$ G* s$^{-1}$. These simulations exposed a second shorter time constant, and we were able to fit the responses accurately by holding that shorter time constant fixed at $\tau_1 = v_{G*}^{-1} = 0.8$ ms. The least-squares fitting procedure reported the rate of rise ($v_{E**}$) to be approximately independent of $Q$, with a mean of 416 E** s$^{-1}$, and reported the second time constant to decrease as $Q$ increased, from an initial value of approximately 5 ms at $Q = 1$. The larger of these two delays, $\tau_2$, is likely to explain the recent report of the existence of a longer delay in the activation of rods compared with cones [8].

On the other hand, the relevance of the decrease in time constant $\tau_2$ predicted in figure 4b with increasing $Q$ is unclear. One might anticipate that this would cause a shortening of the effective delay time $t_{eff}$ required to fit the delayed Gaussian description [2] to the electrical responses. However, for suction pipette recordings from mammalian rods we are not aware of any study where this fit has been examined at intensities as high as 25 000 R* per rod, corresponding to $Q = 20$. And although ERG scotopic $a$-wave experiments have been conducted at these and higher intensities, there is currently controversy regarding the origin of the ERG signal with very intense flashes. Thus, Robson & Frishman [23] report simulations showing that with very bright flashes the ERG $a$-wave includes a substantial signal arising from the flow of capacitive current in the outer nuclear layer, with the consequence that it may be quite inappropriate to attempt to fit and interpret the delayed Gaussian description in the case of ERG $a$-wave responses at very high intensities.

## 4.2. Predicted bright-flash response recoveries and time spent in saturation

The simulated electrical responses in figure 9b (red traces, including aberrant events) provide a close qualitative resemblance to families of flash responses that have been reported in the literature for mammalian rods, in those studies that delivered sufficiently high intensities and/or examined the

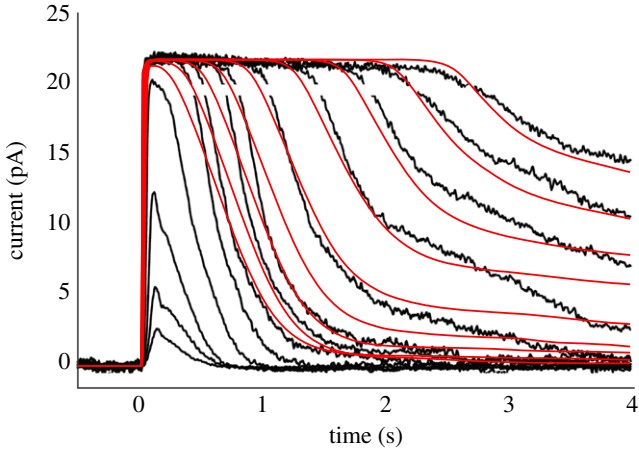

**Figure 12.** Comparison of experimentally recorded family of flash responses from a mouse rod (black traces), from fig. 4A of Burns & Pugh [22], with the predictions of our model for bright flashes (red traces). The estimated flash intensities were extracted from examination of fig. 4B in [22], and may be subject to error. The lowest four intensities were sub-saturating, and have not been modelled; the nine saturating intensities were estimated to range from 330 to 48 000 R* per flash. All parameters were as listed in tables 1 and 2, together with the parameters for aberrant events given in figure 9; at least 10 simulated responses were averaged for each trace. The gaps in the black traces occurred where symbols (shown as red in panel E of electronic supplementary material, figure S1) were removed.

responses out to sufficiently long times. In electronic supplementary material, figure S1 we have collected example flash response families for suction pipette recordings from six such studies, in which the species examined were human [24], monkey [15,16], rabbit [25] and mouse [13,22]. Examination of these panels shows that the averaged intense flash responses typically display a phase of rapid recovery as soon as the current escapes from saturation, followed by a slower phase of decline at later times, and this slower phase becomes more prominent at higher intensities. We are not aware of previous quantitative accounts of this late slower phase, and in §4.4 below we discuss the model that we developed here.

For one of those studies (for a mouse rod in fig. 4A of [22]), figure 12 compares the experimental responses (black traces) with the predictions of our model (red traces). Clearly there is a qualitative similarity between experiment and model, though there are quantitative differences. In considering this comparison, it is important to realize that we were working from published images, and that we did not have the original data or the original flash intensities. We estimated the intensities by examining the semi-logarithmic plot against intensity in fig. 4B of [22], which was for collected cells; hence there are inevitably uncertainties in the comparison, especially in relation to the values of the flash intensities.

Two shortcomings of the model predictions are apparent. Firstly, the initial slopes of the predicted recoveries for the first four saturating flash intensities are lower than seen in the experiment, and the reasons for this deserve future examination. Secondly, the final tail of the predicted recovery after intense flashes is shallower than experiment, presumably indicating that for this mouse rod the mean duration of the aberrant R* shut-off events was shorter than the value of 4 s used in the model. However, given that we did not explicitly alter any parameters in order to describe the kinetics

of this particular mouse rod's responses, we suggest that the level of qualitative agreement is reasonable.

The standard procedure for quantifying bright-flash recovery behaviour is to plot the time $T_{sat}$ that the response remains in saturation as a function of the logarithm of flash intensity [17], which is equivalent to plotting the intensity scale logarithmically. This coordinate system is chosen on the basis that, if an activating substance is produced in direct proportion to light intensity, and if it decays exponentially, then the time spent in saturation should be proportional to the logarithm of flash intensity, and the slope of that line will give the time constant of decay [17]. However, if there are mechanistic departures from proportionality, or if the experimentally observed relationship deviates from a straight line, then the interpretation is not so simple.

The $T_{sat}$ versus $\Phi$ relationships determined from our simulations were presented in figure 11, using the model of dimeric PDE6 activation, either on its own (blue curve), or also taking account of aberrant R* shut-off events (red curve). Because our modelling of the aberrant events is preliminary, we chose parameter values for the model that generated a blue curve marginally shallower than measurements from the literature, together with a red curve marginally steeper.

Both curves display a degree of curvature over the entire range of intensities, so that there is no region that is strictly 'straight line', though it is possible to draw straight lines that provide approximate fits in two regions, as has been done in the literature for measurements from experiments. The dotted straight lines that we have drawn in figure 11 have slopes of $\tau_{D1} = 245$ ms and $\tau_{D2} = 780$ ms, representing the first and second so-called 'dominant time constants' of bright-flash recovery. However, it is important to note that neither of these values corresponds to the time constant of removal of an active substance in the molecular model that we have simulated, though 245 ms differs by only about 20% from the time constant of E** decay, $\tau_{E^{**}} = k_{E^{**}}^{-1} = 200$ ms.

This discrepancy turns out to be accounted for accurately by the super-linearity in E** level that is inherent in the dimeric activation model. Our measurements of the $E^{**}(t)$ tails in figure 7b showed that this super-linearity can be approximated by the expression in equation (5.16), with a fitted 'super-linearity parameter' of $\rho = 0.32$. That relation leads in turn to the prediction that the time $T_{sat}$ for recovery to a fixed criterion level will depend on flash intensity $\Phi$ according to equation (5.18). This predicted relationship is plotted by the heavy dotted black curve in figure 11 and provides a very good description of the blue and red curves in the low-intensity region. Importantly, this predicted relationship is curved, and *not* a straight line. The vertical offset of this curve is set by the horizontal position of the arrow in figure 7b, where the $E^{**}(t)$ trace for a single photoisomerization crosses the saturation level of 2.6 E** per surface at a time of 585 ms.

Furthermore, the slope of the tangent to the curve is given by equation (5.19) as $(1 + (\rho\Phi/N_{surfs}))/k_{E^{**}}$. Substituting $N_{surfs} = 1320$ and $\rho = 0.32$, and at an intensity of $\Phi = 1000$ isomerizations per rod, the factor $\rho\Phi/N_{surfs} = 0.24$, indicating that the slope of the tangent line at this intensity is predicted to be 24% greater than the shut-off time constant of 200 ms, and this is very close to the slope of the shallower dotted line. We therefore have a quantitative rationalization for why the apparent 'dominant time constant' at this level of flash intensity differs from the time constant for shut-off of E**.

royalsocietypublishing.org/journal/rsob Open Biol. **10**: 190241

Published experiments on wild-type mouse rods show a comparable disparity, as exemplified by the measurements tabulated in two studies [12,13] that each analysed around 30 WT rods, and gave the mean ± s.e.m. for both $\tau_{D1}$ and $\tau_{rec}$. Thus, table 1 of Krispel *et al.* [12] gave $\tau_{D1} = 246 \pm 13$ ms ($n = 29$) and $\tau_{rec} = 190 \pm 9$ ms ($n = 33$). Similarly, table 2 of Sakurai *et al.* [13] gave $\tau_{D1} = 271 \pm 13$ ms ($n = 26$) and $\tau_{rec} = 223 \pm 9$ ms ($n = 36$). Application of Welch's *t*-test indicates that we can reject the null hypothesis that the two means are equal, at $p < 0.002$, in both studies. The ratio of the means is $\tau_{D1}/\tau_{rec} \approx 1.29$ in [12] and $\tau_{D1}/\tau_{rec} \approx 1.22$ in [13], bracketing the prediction of our modelling.

For the steeper region of the $T_{sat}$ curve, which has been interpreted as a 'second dominant time constant', it is more difficult to account for the slope, of approximately 780 ms for method 3 or approximately 720 ms for method 2. The slope in this region is bound to be affected by the choice of the parameter $k_{G*}$, representing the rate constant of hydrolysis of free G* which we set to 1 s$^{-1}$, similar to the value reported in [26]. However, the slope obtained will also be affected by non-linearities, including depletion of transducin, and in the case of method 3 by the presence of the late slow phase of response recovery. So, there seems little justification for imagining that this slope directly represents the time constant of any physical process dominating the shut-off of activation.

## 4.3. Parameter values required to explain the observed behaviour

A very important aspect of the predicted $T_{sat}$ versus $\Phi$ relationship, that we have not discussed so far, is the horizontal position of the steeper part of the curve; this can be characterized by the intensity at the intercept of the two fitted straight lines, which we refer to as the transition intensity, $\Phi_{trans}$. In the rods of wild-type mice, this transition intensity has been measured to be approximately 4000–5000 R* per rod [12,13,22,27,28]. Accordingly, we have chosen parameter values that have enabled our simulated relationship to approximately mimic this transition intensity.

It has previously been proposed [28] that the transition intensity corresponds to the situation where all of the PDE6 is 'just covered' by G*, so that any additionally produced G* will remain unbound, and therefore be inactivated at the slower rate constant $k_{G*}$. Our analysis is consistent with this notion, though our quantitative considerations invoke quite different numbers in the calculation of transition intensity than have been used previously. In order to achieve $\Phi_{trans} \leq 5000$ R* we found it necessary to adopt a transducin activation rate of $v_{G*} > 1000$ G* s$^{-1}$ per R*, when other parameters were set to realistic values.

Chief among these other relevant parameter values is the total complement of PDE6, expressed either per disc surface or per outer segment. For a PDE6 membrane density of $C_E = 80$ μm$^{-2}$ (i.e. approx. 1/300 to rhodopsin [29–31]) and an outer segment diameter of $d = 1.3$ μm [18,19], the number of PDE6 holomers per disc surface is $E_{tot} \approx 106$. Then, with the number of disc surfaces $N_{surfs} = 1320$ (for $L = 22$ μm, and 30 discs μm$^{-1}$ [18–21]), the total number of PDE6 holomers per outer segment is 140 000, so that there are 280 000 binding sites for G*.

To cover 280 000 G* binding sites, in response to a flash of 5000 R*, would require an absolute minimum of $280/5 =$

56 G*s produced per R*, though this is an oversimplification as it ignores any shut-off of G* throughout the period of response saturation. For a mean R* lifetime of $T_{R*} = 68$ ms, the necessary rate of G* creation must therefore exceed 56/0.068 s$^{-1}$, or $v_{G*} > 823$ G* s$^{-1}$ per R*. However, given the reality of the occurrence of GTPase activity during the response, the actual requirement must be higher than this. Furthermore, if the mean R* lifetime were instead shorter, at 40–50 ms [32,33], then the required rate would be correspondingly higher.

In our simulations, we used $T_{R*} = 68$ ms. Then, with 140 000 PDE6 holomers per outer segment and $v_{G*} = 1250$ G* s$^{-1}$ per R*, we obtained the fit shown in figure 11. With either a lower rate of transducin activation or a larger number of PDE6 holomers, the steeper segment of the curve was shifted rightward, giving a transition intensity higher than 5000 R*.

Using our standard values of $v_{G*} = 1250$ G* s$^{-1}$ per R* and $T_{R*} = 68$ ms, we obtained the mean number of doubly-activated PDE6 molecules at the peak of the single-photon response as 23.8 E** per R* (figure 2), corresponding to a mean of 47.6 G* per R* bound to PDE6. For comparison, Yue *et al.* [34] have recently presented their analyses of recordings from GCAPs$^{-/-}$ mouse rods, that they interpreted to show a single-photon response of '~12–14 $G_T*\cdot$PDE*s produced per Rho*'; this is three- to fourfold lower than our result, and more in line with values obtained by previous modelling using the earlier estimates of 300–350 G* s$^{-1}$ for the rate of transducin activation. However, as subsequently pointed out by Heck *et al.* [35], the estimate of Yue *et al.* may need (a) to be doubled because it did not consider the possibility that the active state corresponds to the binding of two transducins, and then (b) possibly doubled again because of a substantial discrepancy between the variance and squared mean response traces in their noise analysis [35]. Hence, we do not consider that their estimate differs reliably from ours.

In the downstream cytoplasmic reactions, we obtained a good description of the mean single-photon response, as well as the family of bright-flash responses, using a PDE6 hydrolytic activity parameter of $\beta_{E**} = 0.017$ s$^{-1}$. For comparison, the hydrolytic activity can be predicted from the measured biochemical parameters and physical parameters, as derived in [2,36], to be

$$\beta_{E**} = \frac{k_{cat}/K_m}{N_{Av} \, V_{cyto} \, B_{cG}}, \tag{4.1}$$

where $k_{cat}$ and $K_m$ are the catalytic activity and Michaelis constant for the fully activated PDE** holomers, $N_{Av}$ is Avogadro's number, $V_{cyto}$ is the cytoplasmic volume of the outer segment and $B_{cG}$ is the cytoplasmic buffering power for cGMP. If we adopt values of $k_{cat} = 2750$ s$^{-1}$ [10], $K_m = 10$ μM [37], $V_{cyto} = 0.0146$ pL (half the outer segment envelope volume) and $B_{cG} = 1.6$ (see §5.12.1), then we obtain a predicted value of $\beta_{E**} = 0.0195$ s$^{-1}$, which represents as close agreement to our chosen value as we could reasonably expect.

## 4.4. Late slow phase of recovery

In this paper we have shown that a plausible description of the late phase of recovery can be achieved by taking account of aberrant R* shut-off events, which have been clearly demonstrated to occur at lower intensities, of a few hundred

to a few thousand isomerizations in monkey rods [15,16], and of several tens of isomerizations in GCAPs$^{-/-}$ mouse rods [6]. We used values for the frequency of occurrence and mean event duration, of $p_{aberr} = 0.002$ and $\tau_{aberr} = 4$ s, close to those reported in the literature, and we set the plateau amplitude of an individual aberrant event to be approximately equal to the amplitude of the normal single-photon response, at 4–5% of the dark current (see below). Our simulations of the photoresponses that result from the combination of normal R* shut-off events and a very low rate of aberrant events (figures 9 and 10) demonstrated a late phase of response recovery that appears broadly consistent with experiment. This regime of flash intensities and recovery times has been examined in relatively few studies (examples of which are presented in electronic supplementary material, figure S1), but the phenomenon has not been investigated comprehensively, probably because of the excessive time that is needed to elapse between presentation of repeated very intense flashes. In our view, the simulated mean responses in the red traces of figure 9b reproduce the qualitative features of the late phase of recovery quite well, as shown for one cell in figure 12, and we likewise consider that the examples of simulated raw responses in figure 10 show fluctuations consistent with experiment. However, there is a paucity of relevant experimental data in the literature, and we acknowledge that our present description is preliminary.

It should be noted that our choice to assign the aberrant events a similar plateau amplitude to the peak of the normal single-photon event differs from the view in the literature that the aberrant events are equivalent to single-photon events in rods of GRK1$^{-/-}$ animals (i.e. that lack any shut-off of R*); thus R* events in GRK1$^{-/-}$ animals typically have a plateau of more than double the amplitude of the events in WT animals. Our reasons for this choice included the following. Firstly, Kraft & Schnapf [16] reported a plateau level of approximately 0.8 pA for the aberrant events in monkey rods, similar to the peak of the SPR. Secondly, when we instead adopted the 'GRK1$^{-/-}$ model', the magnitude of the fluctuations in our simulations of individual raw responses (comparable to those in figure 10) appeared excessively large. Thirdly, we are not aware of any firm evidence that during the aberrant R* shut-off events the catalytic activity of R* remains unaltered. But, finally, we note that the assumed fractional activity ($a_{aberr}$) of the aberrants events does not affect the qualitative form of the late stage of recovery of the mean electrical response.

## 4.5. Implications of the model for phototransduction in rods with abnormal genes

Abnormalities of phototransduction, that are caused either by naturally occurring or genetically engineered mutations, are of significant interest to photoreceptor physiologists and retinal disease researchers. Of particular significance to our present analysis will be mutations of the genes encoding PDE6, PDEγ, and transducin, because these would seem to offer the greatest potential for modifying the dimeric nature of PDE activation modelled here. For example, if it were possible to replace the heterodimeric PDE6αβ with a homodimeric PDE6αα or PDE6ββ, or alternatively to express a chimeric form of the kind studied by Muradov et al. [38], then it might be possible either to reduce or to eliminate the native super-linearity of the PDE6, and to examine the consequences

of this. Likewise, it would be interesting to mutate certain residues of PDEγ to those found in the cone protein (see, for example, [39]), and to examine whether the activation of the PDE6 became more nearly linear. Similarly, it would be of considerable interest to use the model presented here to analyse the responses of rods exhibiting altered interaction between transducin and PDEγ, as occurs with the W70A mutation of PDEγ [40], or in rods with lowered efficacy of transducin activation [34].

Mutations in a protein of interest may result in clear and straightforward changes in the photoreceptor's response, provided that the expression levels of other proteins are unaltered, but it is often the case that the expression levels of other proteins are indeed modified, leading to a degree of homeostasis in the cell, and hence to difficulty in interpretation of the underlying molecular mechanisms. We think that the model developed here may be of use to physiologists in attempting to separate effects that are directly attributable to a known mutation from the homeostatic responses of the cell to such a mutation. Thus, the model presented here may be a useful tool for dissecting apart the primary and secondary events caused by these molecular modifications.

## 4.6. Summary and future directions

We have presented an updated model of rod phototransduction, based on the recent evidence that activation of the dimeric PDE6 requires the binding of two molecules of transducin [10], and we have evaluated the model's predictions for the form of the rod's electrical response to bright flashes of light. The predicted responses closely mimic experiment, and our new description eliminates a number of shortcomings of previous models (that we enumerated in the Introduction). Of particular note, the transition intensity for the change of slope in semi-logarithmic plots of saturation time $T_{sat}$ versus flash intensity $\Phi$ is correctly obtained as $\Phi_{trans} \approx 5000$ photoisomerizations per flash (figure 11). The value we adopted for the rate of transducin activation, of $\nu_{G*} = 1250$ G* s$^{-1}$ per R*, is consistent with light-scattering measurements [41–44], as discussed recently in [10,11]. Likewise, the value we adopted for the hydrolytic activity of the doubly activated PDE6, of $\beta_{E**} = 0.017$ s$^{-1}$, is close to the value predicted from biochemical measurements and physical factors.

Even though our rate of transducin activation is around fourfold higher than assumed by others, this rate can be put into perspective by calculating the numbers of molecules involved during the single-photon response. Over the lifetime of an individual R* molecule, our simulations give the total number of transducin molecules activated as 84 G*, with the great majority of these (approx. 83) having been created by the time of the peak of the mean single-photon electrical response at 130 ms. At that time, the mean number of doubly activated PDE6 molecules is approximately 24 E** (figure 2), so that approximately 47 transducins are bound in the fully activated form. In addition, another approximately 17 transducins are singly bound, while just 2–3 are unbound, and the remaining approximately 16 have already been inactivated by GTPase activity during the first 130 ms. Thus, the combination of a high rate of transducin activation with a short mean R* lifetime (approx. 68 ms) leads to quite modest numbers of activated molecules on the disc membrane during the single-photon response.

For the future, we intend to obtain more extensive bright-flash response measurements from mammalian rods, so as to test the predictions more comprehensively and to extract estimates of parameter values across species. We anticipate that examination of responses from rods expressing mutant forms of PDE6, PDEγ or Gα may help in specifying these parameters. There is also scope for more exhaustive tests of whether the late slow phase of recovery is accounted for by aberrant R* shut-off events, and in this case the use of mutations in the C-terminal region of rhodopsin may be valuable. In parallel, we plan to investigate whether we can develop improved numerical approaches that provide appreciably shorter computation times for simulating the responses, because the approaches we have presented here are very time-consuming, and not readily amenable to fine-tuning the values of parameters. One such potential approach is foreshadowed in §5.7. We also intend to examine whether we can extend the model to cone photoreceptors, with their much faster, less-sensitive and noisier responses. And, in light of the structural information (including high resolution cryo-EM) that is now available for the PDE6 (e.g. [10,45,46]), we hope to investigate the link between molecular structure and state of PDE activation.

# 5. Methods and theory

This section describes the simulation methods that we used to obtain the bright-flash responses. In addition, it presents the analytical solutions that we derived in some simplifying situations. And it also describes several checks and other investigations that we conducted.

## 5.1. Method 1: Full 2-D simulation of diffusional interactions at the disc surface

For sub-saturating flash intensities, it is rare for any disc surface to receive more than a single photoisomerization. For example, at a just-saturating intensity of $\Phi = 150$ R* per rod, and assuming approximately 1300 disc surfaces per rod, the mean number of isomerizations per surface is approximately 0.1, and it can be shown that fewer than 1% of the surfaces experience multiple isomerizations. Hence, over much of the rod's operational intensity range up to saturation, its electrical response can be calculated from knowledge of the effect of a single photoisomerization per disc surface.

In recent analyses [11,14], we have modelled this regime, of a single photoisomerization per disc surface, and we now briefly recapitulate. We modelled the shut-off of R* activity in terms of multiple steps of phosphorylation, together with the assumption that the drop in rhodopsin's enzymatic activity results not from phosphorylation *per se*, but instead primarily from the binding of arrestin, for which the probability of binding increases dramatically once several (e.g. 3) phosphates have attached [14]. We modelled the interactions of proteins at the disc surface using numerical simulation of 2-D diffusion, with the 'shortcut' simplification that transducin is activated to G* stochastically, at a fixed mean rate while R* remains active [47]. The activated G*s then diffuse laterally (from wherever they were created), and are able to bind to PDE6, either in its unbound form (denoted E) or in its singly bound form (E*), thereby creating either E* or the doubly bound form, E**, respectively. The shut-off of E* and

E** is assumed to occur stochastically, as a result of GTPase activity, returning the molecule to its resting (E) or singly bound (E*) state, respectively. A full description of our molecular model was set out in [11]. Using estimated values for the parameters, we numerically evaluated its predictions and showed good agreement with experiment in the sub-saturating intensity range. In that earlier work we ignored any decay of unbound G* (which is reasonable in the case of a single isomerization per surface). Here, though, we instead allow unbound G* to shut off stochastically with a rate constant $k_{G*}$ close to that reported in the literature [26], because this pathway has been proposed to make a major contribution at very high intensities when all the PDE6 binding sites have G* bound [28].

We refer to this approach using numerical simulation of the 2 D lateral diffusion reactions at the disc surface as method 1. Our Matlab code for all three simulations methods, and also for numerical integration of the downstream reactions, is available for download (see Data accessibility).

## 5.2. Equations describing rising phase of PDE6 activity with multiple isomerizations per disc surface

In previous work, for the case of a single photoisomerization ($Q = 1$), we fitted the onset phase of $E^{**}(t)$ with the expression for a ramp in time convolved with an exponential decay in time, representing a single delay stage of roughly 7 ms that we predicted should precede the ramp-wise appearance of the doubly activated E**; see eqn (2.1) of [11]. Here, though, with responses to multiple R*s per disc surface, it became apparent that a second shorter delay stage is additionally required, to provide an adequate description at very early times. Furthermore, we discovered that by fixing this shorter time constant at the expected first-contact time for activation of G* by R*, namely at $\tau_1 = 1/\nu_{G*}$, we could achieve a very good fit to the rising phase of the entire set of $E^{**}(t)$ responses, as indicated by the dotted traces in figure 3*b*.

If we take a ramp in time, $E^{**}(t) = Q \nu_{E^{**}} t$ for $t > 0$, where $\nu_{E^{**}}$ denotes the rate of E** activation per isomerization, and we convolve this with two exponential decay stages having time constants $\tau_1$ and $\tau_2$ (for $\tau_1 \neq \tau_2$), we obtain the doubly delayed ramp expression

$$E^{**}(t) = Q \nu_{E^{**}} \{t - (\tau_1 + \tau_2) \\ + (\tau_1^2 e^{-t/\tau_1} - \tau_2^2 e^{-t/\tau_2})/(\tau_1 - \tau_2)\}, \quad t > 0. \quad (5.1)$$

In fitting this equation to the simulations, our procedure was to set $\tau_1 = 1/\nu_{G*}$, and then find the values of $\nu_{E^{**}}$ and $\tau_2$ that provided the least-squares best fit to the rising phase for each mean response, within the region indicated by the dashed curve in figure 3*b*, corresponding to an ellipse with radii 30 ms and $0.35\ E_{tot}$. Inspection of the traces in figure 3*b* shows that each individual dashed curve provides an excellent description of the data over the fitted region.

## 5.3. Differential equations for the case of spatial homogeneity

In order to solve for responses to bright flashes, with multiple R*s per surface, the only approach available for the onset phase of the response is method 1 using numerical simulation of 2D diffusion. However, for the recovery phase of the response, requiring simulation to times of perhaps 10 s, this approach

becomes exceedingly time-consuming computationally, and we therefore developed a much faster approximate method, that we term method 2. This approach uses a mass-action approximation, and should be valid once the distribution of interacting molecules (G*, E, E* and E**) has relaxed to approximate spatial homogeneity (see below). The approach also requires that R* activity has essentially ceased, so that there is no longer any punctate creation of G*. Both of these requirements will be satisfied within a few hundred milliseconds of flash delivery, for the parameters chosen in our model. And after that time, we simply model the massed interactions between the molecules in the reaction scheme shown in figure 1. We now derive the differential equations that underlie this approximate approach.

As in model 1, in this spatially homogeneous model we assume that all lateral interactions occur at their diffusion limit. These lateral diffusional contacts between two molecular species may be considered in terms of the average of the 'sweeping out' of surface area by individual molecules. For example, a single G* molecule will 'sweep out' molecules of E that are uniformly distributed across the disc, in proportion to the spatial density of E. More generally, the rate of contacts by a single molecule of one species is given by the sum of the two lateral diffusion coefficients multiplied by the area density of the other species. Then, to obtain the average rate across all molecules of the first species, we simply multiply by its spatial density. In the terminology of figure 1, we can write the rate at which molecules of G* contact molecules of E as

$$r_1(t) = k_1 \, G^*(t) \, E(t), \tag{5.2}$$

where the appropriate bimolecular rate constant is $k_1 = (D_{G^*} + D_E)/A$. Likewise, for the second step we can write the rate at which molecules of G* contact molecules of E* as

$$r_2(t) = k_2 \, G^*(t) \, E^*(t), \tag{5.3}$$

where the appropriate bimolecular rate constant is $k_2 = (D_{G^*} + D_{E^*})/A$. In practice, $k_1$ and $k_2$ will be very similar to each other.

The differential equations for transducin and the three forms of PDE6 may then be written as

$$\frac{d}{dt}E(t) = -r_1(t) + k_{E^*} \, E^*(t), \tag{5.4}$$

$$\frac{d}{dt}E^*(t) = r_1(t) - k_{E^*} E^*(t) - r_2(t) + k_{E^{**}} E^{**}(t), \tag{5.5}$$

$$\frac{d}{dt}E^{**}(t) = r_2(t) - k_{E^{**}} E^{**}(t) \tag{5.6}$$

and $\quad \dfrac{d}{dt}G^*(t) = -r_1(t) - r_2(t) - k_{G^*} G^*(t). \tag{5.7}$

Note that the five rate parameters in the equations above ($k_1$, $k_2$, $k_{E^*}$, $k_{E^{**}}$ and $k_{G^*}$) are constants. Finally, we have the conservation relation for PDE6 that

$$E(t) + E^*(t) + E^{**}(t) = E_{tot} \tag{5.8}$$

so that any one of equations (5.4)–(5.7) may be omitted, by substitution.

Numerical solution of equations (5.2)–(5.8) forms the foundation of method 2, because we have not been able to derive an analytical solution of the equations as they stand. Nevertheless, we are primarily interested in cases where $G^*(0)$ is large, so that $E^*(0)$ is small and $E(0)$ is even smaller. Hence, we have the approximation that $E^{**}(0) \approx E_{tot}$, and in

this limiting case an analytical solution is straightforward, as we derive in §5.5.

### 5.3.1. Time for the attainment of spatial homogeneity

The punctate activity of R* molecules will have substantially ceased within approximately 100 ms of flash delivery (see R*(t) trace in figure 2a), and thereafter the spatial distribution of reactant molecules will relax towards spatial uniformity. The time constant of this equilibration should be given approximately by the relevant surface area divided by the lateral diffusion coefficient $D$ for the molecular species. Following the delivery of $Q$ isomerizations randomly across the disc surface, the relevant area will be the total disc surface area divided by $Q$. Hence the effective spatial equilibration time will be $T_{equil} \approx A_{disc}/(Q \, D)$. For an outer segment diameter of 1.3 μm, the surface area is $A_{disc} = 1.33 \, \mu m^2$. For $Q = 5$ R* per surface (the minimum intensity for which the PDE is approximately saturated) and for transducin, with $D_{G^*} = 2.2 \, \mu m^2 \, s^{-1}$, the effective spatial equilibration time will therefore be $T_{equil} \approx 120$ ms; for $Q = 10$ R* per surface, the effective equilibration time will drop to 60 ms. Hence, we can expect that within a few hundred milliseconds of the delivery of 10 R* per surface the assumption of spatial homogeneity should be valid.

## 5.4. Stochastic fluctuations contributing to form of bright-flash responses

As foreshadowed in §2.3, it is perhaps counterintuitive that in the case of very bright flashes it turns out to be important to take account of two sources of stochastic fluctuations: namely, fluctuations in quantal absorptions per disc surface, and fluctuations in R* lifetime.

### 5.4.1. Stochastic fluctuations in number of photoisomerizations per disc surface

For a rod outer segment with $N_{surfs}$ disc surfaces, a flash that delivers a total of $\Phi$ photoisomerizations will elicit a mean number of $\phi = \Phi/N_{surfs}$ photoisomerizations per disc surface. The Poisson distribution then gives the probability $p_Q$ that a disc receives $Q$ (an integer) isomerizations as

$$p_Q = \frac{\phi^Q}{Q!}\exp(-\phi). \tag{5.9}$$

Our approach below is to consider all integer values of $Q$ up to some suitable limit (say, 60, when $\phi = 30$), and to consider separately the solution in each subset of disc surfaces receiving that number of isomerizations.

### 5.4.2. Stochastic fluctuations in R* lifetime

For the set of disc surfaces that each experience the same number, $Q$, of isomerizations, there will nevertheless be differences in the extent of activation, arising from stochastic fluctuations in the lifetimes of the individual R* molecules. For a numerical solution, we can simulate the $Q$ individual R* lifetimes, and this is one approach that we take, though it is computationally time-consuming. For a faster (semi-analytical) approach, we note that all these isomerizations occur synchronously, as we are modelling a very brief flash. Then, by making the assumption that each R* activates molecules

of transducin independently, we can approximate the effect of the multiple isomerizations in terms of the 'total R* activity' summed across the $Q$ molecules of R*. Specifically, we calculate the expected distribution for the sum of the $Q$ stochastic lifetimes, and we take this distribution of total activity and we assume that it applies over a short time interval (corresponding to the mean R* lifetime). This approach is valid because the mean R* lifetime (68 ms) is very short in comparison with the response times being examined for bright flashes.

To do this, we begin with the distribution of lifetimes for a single activated R*. In our recent re-analysis of rod single-photon responses, we developed a model of 'binary shut-off' of R* activity, wherein R*'s activity was assumed to be constant until it dropped to zero upon arrestin binding, which occurred stochastically after the binding of $M$ phosphates. We were able to obtain a good description of experimental measurements of single-photon responses by making the simplest assumption that each of the rate constants of the $M + 1$ reactions ($M$ phosphates plus arrestin binding) were equal, at $\mu = 60 \text{ s}^{-1}$. With all rate constants equal, the kinetics are often referred to as 'Poisson' (see [48]), though they should perhaps more accurately be referred to as following a gamma distribution. In this case, the probability density function (pdf) of R* lifetimes is given by

$$\text{pdf}_1 = \frac{\mu(\mu t)^M \text{e}^{-\mu t}}{M!}, \tag{5.10}$$

and the time integral of this expression gives the cumulative probability function (cdf) as

$$\text{cdf}_1 = 1 - \text{e}^{-\mu t} \sum_{k=0}^{M} \frac{(\mu t)^k}{k!} = 1 - \overline{R^*(t)}, \tag{5.11}$$

where the right-hand side of equation (5.11) links the expression to the mean time-course of R* activity that was derived as eqn (2.3) in [14]. In both equations above we have employed a subscript '1' to denote the case of a single photo-isomerization. The cdf in equation (5.11) may also be expressed as the incomplete gamma function, $\gamma(M + 1, \mu t)$, with integer shape parameter $M + 1$ and with rate parameter $\mu$ (see e.g. Abramowitz & Stegun [49], §6.5.13).

For our purposes, a crucial property of the gamma distribution is that the sum of $Q$ independent samples (with common second parameter) is also gamma distributed, with its shape parameter summed across the $Q$ underlying distributions. Thus, for $Q$ isomerizations each with shape parameter $M + 1$, we have

$$\text{cdf}_Q = \gamma(Q(M + 1), \mu t) \tag{5.12}$$

so that the pdf may be written as

$$\text{pdf}_Q = \mu(\mu t)^{Q(M+1)-1} \text{e}^{-\mu t}/(Q(M + 1) - 1)! \tag{5.13}$$

This last expression gives the probability density, across trials, for the summed activity of the $Q$ isomerizations that were delivered to the disc surface at time zero. When the exponent $Q(M + 1) - 1$ is very large (e.g. 119, for $Q = 30$ isomerizations per surface and $M = 3$ sites required to be phosphorylated for arrestin binding), evaluation of equations (5.12) and (5.13) can become problematic, because of the combination of the large exponent and division by the large factorial. However, evaluation is straightforward in Matlab, using the functions gammainc and poisspdf in the statistics toolbox.

## 5.5. Method 2: Numerical simulation of the spatially-homogeneous macroscopic approximation for bright-flash recoveries

Based on the theoretical considerations above, together with our analysis of depletion of G-protein at extremely high intensities (§5.9), we implemented a composite approach (termed method 2) to approximating the recovery phase of the bright-flash response, that was orders of magnitude faster than the full 2-D diffusion simulations of method 1, yet that conformed closely with those very slow simulations.

For each fixed number $Q$ of isomerizations per disc surface, in the range of interest (e.g. 1 … 60), we ran a substantial number of iterations (typically 1000) of the following procedure. We generated $Q$ pseudorandom R* lifetimes, as in method 1, and described in detail in [14]. Then we numerically solved the set of ordinary differential equations comprising generation of G* at rate $v_{G^*}$ (with allowance for G-protein depletion) for the set of $Q$ R* molecules with these lifetimes, in conjunction with recovery according to numerical integration of equations (5.2)–(5.8).

However, because spatial homogeneity was unlikely to have been achieved for several hundred milliseconds, as a result of the punctate R* activity prior to its quenching and the subsequent time required for spatial equilibration, it was necessary to modify these equations during the initial post-flash period. We found that, by reducing the first bimolecular rate constant ($k_1$) by a factor of 5-fold until 400 ms after the flash, the simulated mean activities for method 2 agreed closely with the results obtained by the much more time-consuming method 1. We do not have a theoretical explanation of why this particular adjustment worked, but clearly it was necessary to make some kind of modification during the initial period prior to the attainment of spatial homogeneity, and in practice this adjustment was found to work, in the sense that the results of method 2 then conformed closely with those of method 1.

We found that method 2 was more than 2000-fold faster than method 1; thus, with $Q = 10$, method 1 took approximately 17 h for 200 repetitions (approx. 5 min per trial), whereas method 2 took just 2 min for 1000 repetitions (approx. 120 ms per trial). Nevertheless, repeating those method 2 calculations across the 60 values of $Q$ took around 2 h.

## 5.6. Analytical approximation for the case of spatial homogeneity, as a check on the numerical simulations in method 2

Although we were not able to obtain a general solution to the set of differential equations in equations (5.2)–(5.8), there is a useful simplification that applies over much of the time-course of interest, when sufficient G* remains present to react rapidly with any E or E* that is formed by the shut-off reactions. And that is to assume that, throughout this period, there is rapid equilibration between E* and E** with the level of unbound E remaining negligible (i.e. $E(t) = 0$). This means that, in figure 1, the forward flux $r_1$ is zero. Hence $E^*(t)$ is small, and as a result $E^{**}(t) \approx E_{\text{tot}}$. Furthermore, under these conditions the forward flux $r_2$ will equal the reverse flux (i.e. $r_2(t) = k_{E^{**}} E^{**} \approx k_{E^{**}} E_{\text{tot}}$). Substituting these expressions for $r_1$ and $r_2$, equation (5.7) simplifies to

$$\frac{\text{d}}{\text{d}t} G^*(t) = -k_{E^{**}} E_{\text{tot}} - k_{G^*} G^*(t). \tag{5.14}$$

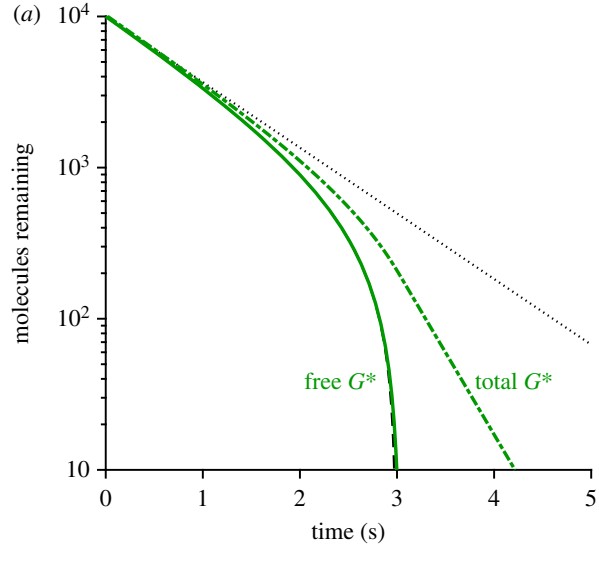

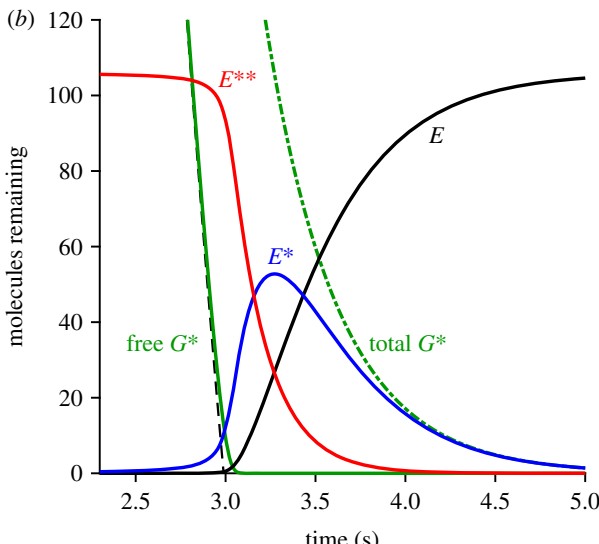

**Figure 13.** Numerical solution of the set of differential equations for the spatially homogeneous case, equations (5.2)–(5.8), from an initial level of $G^*(0) = 10\,000$ activated transducins. All parameters in these equations were set to the values listed in table 1. (a) Semi-logarithmic plot for decay of activated transducin. (b) Comparison with results for transducin and PDE using method 2, in linear coordinates. Free $G^*(t)$ is solid green; total $G^*(t)$ is dot-dash green. $E(t)$ is black; $E^*(t)$ is blue; $E^{**}(t)$ is red. The dashed black trace in both panels plots the analytical approximation in equation (5.15), which provides a predicted lower limit on the numerical solution for free $G^*(t)$. The dotted straight line in A plots an exponential decline with rate constant $k_{G^*}$.

This differential equation has the solution

$$G^*(t) = \left(G^*(0) + \frac{k_{E^{**}} E_{tot}}{k_{G^*}}\right) \exp(-k_{G^*} t) - \frac{k_{E^{**}} E_{tot}}{k_{G^*}}, \qquad (5.15)$$

which provides a convenient check, as a lower limit for our numerical solution for $G^*(t)$.

Figure 13 plots the results of our numerical solution of the full set of differential equations, equations (5.2)–(5.8), for an initial quantity of free activated transducin of $G^*(0) = 10\,000$ molecules, and with the standard set of parameters that we used in the Results section. Panel $a$ of figure 13 is a semi-logarithmic plot for activated transducin: both for its free form, $G^*(t)$ (continuous green), and also for its total

including molecules bound to PDE6, $G^*_{tot}(t)$ (dot-dash green). For comparison, the dashed black curve plots the approximate check in equation (5.15), and is almost hidden by the green curve, while the dotted straight line plots an exponential decline. Figure 13$b$ is in linear coordinates, and additionally plots the numerical solutions for $E(t)$ in black, $E^*(t)$ in blue and $E^{**}(t)$ in red; note the truncated time scale in this second panel.

## 5.7. Potential future shortcut for bright-flash recovery

Although the computation time mentioned above for method 2 was acceptable for our final calculations, it would be very useful to have an even faster approach for future investigations, and so we conducted preliminary tests of a potential further shortcut. For a sufficiently large initial value of activated transducin, say $G^*(0) = 10\,000$, we numerically solved the set of ordinary differential equations (equations (5.2)–(5.8)) just once, to generate $G^*(t)$, $E(t)$, $E^*(t)$ and $E^{**}(t)$, exactly as in figure 13. Then, for each required value of $Q$, we generated the probability distribution predicted by equation (5.13) for the sum of R* lifetimes, and we scaled this by $v_{G^*}$ (again, with allowance for G-protein depletion) to obtain the probability distribution of $G^*(0)$ expected for that value of $Q$. We then interpolated the solved function $G^*(t)$ to find the time-shift corresponding to each such initial value. Then, for each molecular species, we correspondingly time-shifted the waveform, and took a weighted average (weighted according to the calculated probability distribution) to obtain the predicted mean kinetics for that molecular species.

This approach would not be applicable for small numbers of isomerizations (say $Q < 6$) where E** does not reach saturation, and nor would it be applicable at the earliest times, because the predicted initial level of E** is $E^{**}(0) \approx E_{tot}$ rather than zero. Furthermore it requires an additional time offset, to account for the fact that transducin is not activated instantaneously. Despite these shortcomings, we found the approach to be very promising. Thus, we found that for $Q \geq 10$ this method generated predictions that were almost indistinguishable from those obtained with method 2, yet it was a further factor of more than 1000-fold faster. Accordingly, we plan to put the approach onto sounder foundations in a future investigation.

## 5.8. Super-linearity of the PDE6 response

In §3.3 it was shown that the $E^{**}(t)$ activation traces for the simulated bright-flash responses of figure 7$b$ exhibit substantial super-linearity, so that the vertical scaling of the tail-phase increases more steeply than linearly with intensity. In figure 7$c$, the response amplitude $R_Q$ (at a fixed time) was shown to be described by the equation

$$\frac{R_Q}{R_1} = Q(1 + \rho)^{Q-1}, \qquad (5.16)$$

where $Q$ is the number of isomerizations per disc surface, and $\rho = 0.32$ is the super-linearity parameter. We now show that this super-linearity in the simulated responses of the dimeric model of PDE6 activation leads to the prediction of a significant deviation from the notion of a 'dominant time constant', that has been used to characterize the relationship between flash intensity and the time $T_{sat}$ that the electrical response spends in saturation [17].

royalsocietypublishing.org/journal/rsob Open Biol. **10**: 190241

To allow for the stochastic distribution of photoisomerizations, we weight the $R_Q$ amplitudes according to the Poisson probability distribution of $Q$, at any flash intensity $\phi$ (in R* per surface). This gives

$$R_\phi = \exp(-\phi)\left[\phi R_1 + \frac{1}{2}\phi^2 R_2 + \frac{1}{3!}\phi^3 R_3 + \frac{1}{4!}\phi^4 R_4 + \ldots\right]$$

$$= \exp(-\phi)\phi\left[1 + (1+\rho)\phi + \frac{1}{2}((1+\rho)\phi)^2\right.$$

$$\left. + \frac{1}{6}((1+\rho)\phi)^3 + \ldots\right]R_1$$

$$= \exp(-\phi)\phi\exp((1+\rho)\phi)R_1$$

$$= \phi\exp(\rho\phi)R_1. \tag{5.17}$$

In the classical model of independent activation of PDE6 subunits, super-linearity does not occur, and instead we would have $\rho = 0$, so that equation (5.17) would reduce to $R_\phi = \phi R_1$.

In order to calculate the time taken for the tail to recover to any fixed level of E** (e.g. for the time $T_{\text{sat}}$ spent in saturation), we note that in the tail region $R_1 = c\exp(-k_{E^{**}}t)$ where $c$ is a constant, and then we set $R_\phi$ to be constant, so that we obtain the time $T_{\text{sat}}$ to reach some criterion level of $E^{**}(t)$ as

$$T_{\text{sat}} \approx \ln(\phi\,\mathrm{e}^{\rho\phi})/k_{E^{**}} + \ln\left(\frac{c}{R_{\text{sat}}}\right)/k_{E^{**}}. \tag{5.18}$$

The second term in this equation is a constant vertical offset, corresponding to the time at which the $E^{**}(t)$ response to a single photoisomerization per surface ($Q = 1$) declines to the level that just causes saturation, of $R_{\text{sat}} \approx 2.6$ E** per surface. That time is indicated by the arrow in figure 7b, and has a magnitude of 585 ms.

From this analysis of super-linearity in the model of dimeric PDE activation, equation (5.18) predicts that the relationship between $T_{\text{sat}}$ and the logarithm of flash intensity should be *curved* rather than a straight line. If, instead, the PDE were activated in a strictly linear manner, then $\rho$ would be zero, and the first term in equation (5.18) would reduce to $\ln \phi/k_{E^{**}}$, giving the conventional straight-line relationship. In the general case, we can obtain the slope of the tangent line in a semi-logarithmic plot against intensity by differentiating equation (5.18) with respect to $\ln \Phi$ (which is the same as differentiating with respect to $\ln \phi$), to obtain

$$\frac{\mathrm{d}T_{\text{sat}}}{\mathrm{d}\ln\Phi} = \frac{1 + (\rho\Phi/N_{\text{surfs}})}{k_{E^{**}}}. \tag{5.19}$$

This indicates that the apparent 'dominant time constant' will exceed the time constant of E** shut-off by a factor of $\rho\Phi/N_{\text{surfs}}$ (i.e. by a factor of approximately 24% when the tangent is measured at $\Phi = 1000$ R*). The predictions of equations (5.18) and (5.19) were compared with our simulations of the electrical responses and with experiment in §4.2.

## 5.9. Depletion of transducin Gαβγ with very intense flashes

To investigate the depletion of G-protein in the case of very intense flashes, when there are multiple photoisomerizations per disc surface, it is necessary to use a 2D diffusional simulation. However, the WalkMat code of method 1 is not appropriate in its current form, because it employs the

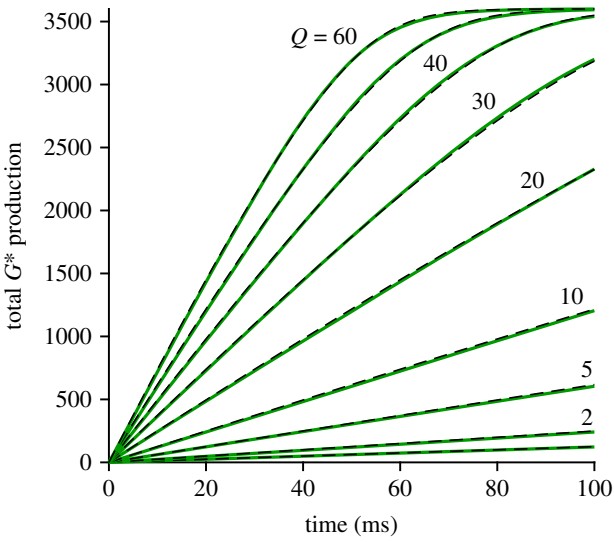

**Figure 14.** Walk2 simulations of G* production. Green traces are the means of 100 simulations in Walk2, for the indicated numbers $Q$ of R* molecules on the disc surface ($Q = 1$ and $Q = 50$ are not labelled). Dashed black traces plot the analytical expression in equation (5.21) with $K_{\text{m}} = 0.14$. The Walk2 program uses a square region for simulation (set to 1.2 μm on a side), and it explicitly includes lateral diffusion of G-protein molecules. Parameters were as in table 1, except that all shut-off reactions were disabled, and the PDE was removed from consideration by setting $C_E = 0$.

'shortcut' approximation that each R* simply activates G*s at a fixed mean rate; re-coding to include diffusion of the individual G-protein molecules (at a 30-fold higher concentration than the PDE) would slow the computations by a further order of magnitude. Instead, we chose to simulate the lateral diffusion of the holomeric G-protein molecules and their interactions with R* molecules using the earlier 'Walk2' program written by Lucian Wischik in 1996 (see Data accessibility). That program has the advantage that the lateral diffusional contacts of each of the molecular species are simulated (rather than using the 'shortcut' activation of G*), yet it executes very rapidly, in part because of its use of a 'quick-and-dirty' random number generator. However, its disadvantages are that multi-step inactivation of R* is not implemented, and the simulated region is square rather than circular.

The (stochastic) reaction time following diffusional contact between an R* and a G was set to 0.8 ms, thereby providing an initial rate of G* activation of $\nu_{G^*} = 1250\,\mathrm{s}^{-1}$; the density of transducin was set to $C_G = 2500\,\mathrm{μm}^{-2}$ (which gave 3600 Gαβγ molecules on the simulated square region of membrane, 1.2 μm on a side). In order to separate depletion effects from shut-off effects, we disabled shut-off of both R* and G*. In addition, we removed PDE (by setting $C_E = 0$), so that the only reactants present initially were R* and G. Accordingly, the simulated trajectories for $G^*(t)$ represent the total numbers of transducin molecules activated, in the absence of all inactivation reactions. The results obtained are presented in figure 14.

The green traces plot the simulated activation of G*, when different numbers $Q$ of R* molecules were present on the disc surface, averaged in each case from 100 repetitions, and with $Q$ as indicated near the traces. Each trace begins ramping upwards, with initial slope $Q\nu_{G^*}$, but the slope gradually decreases as G* approaches the total complement of transducin (i.e. as transducin is depleted). Note that the uppermost

trace (for $Q = 60$ R* per surface) shows almost complete exhaustion of transducin within 60–70 ms of the flash. In these simulations, shut-off of R* was disabled. In the normal case, where multi-step shut-off occurs with a mean lifetime of 68 ms, R* would have been almost fully active for the whole of this period; see, for example the shape of the $R*(t)$ trace in figure 2. Hence, for this $Q = 60$ trace, our elimination of R* shut-off should have made little difference to the predicted time-course of transducin depletion.

Somewhat to our surprise, we found that simply by scaling the time axis in proportion to $Q$, the simulated traces superimposed upon each other. However, in figure 14, rather than scaling the green traces in time, we have instead scaled the theoretical curves. Thus, as shown by the dashed black traces, we found that a common template curve provided a good fit to each of the traces, when scaled in time in proportion to $Q$. The common curve that we found to provide a good fit has the form of a rate-limited saturation, as expected if the rate of transducin depletion is not constant, but instead saturates in a Michaelis–Menten manner, described by

$$\frac{dG}{dt} = -v_0 \frac{G/G_0}{G/G_0 + K_m}(1 + K_m), \qquad (5.20)$$

where $G$ denotes the quantity of G$\alpha\beta\gamma$ remaining, and where $K_m$ is the saturation constant (as a fraction of the initial transducin level, $G_0$). The parameter $v_0$ is the initial rate of depletion (i.e. $v_0 = Q\, v_{G*}$).

From this relation it can be shown that the fractional depletion of transducin, $1 - G(t)/G_0$, will be given by

$$1 - \frac{G(t)}{G_0} = K_m W\left\{\frac{1}{K_m}\exp\left(\frac{1}{K_m} - \left(1 + \frac{1}{K_m}\right)\frac{v_0}{G_0}t\right)\right\}, \qquad (5.21)$$

where $W(x)$ is the Lambert W function [50]. Note that it is the factor of $Q$ built into $v_0$ that accomplishes the time-scaling of the common underlying curve. In fitting equation (5.21) to the simulated responses in figure 14, we found that the appropriate value for the saturation constant was $K_m \approx 0.14$.

The outcome of this analysis is that we found a simple function to describe the depletion of G-protein, that we could use in conjunction with the 'shortcut' method of simulating G* activation. In other words, this analysis confirmed that there is no need to model diffusion of the large number of transducin molecules in our simulations. We were therefore able to apply this description of G-protein depletion to all of our simulations of bright-flash responses.

## 5.10. Method 3: Incorporation of aberrant R* shut-off events

To analyse the PDE6 activity when aberrant R* shut-off events are taken into consideration, we specify that there is a very small probability, $p_{aberr} \approx 0.002$, that an R* fails to inactivate normally, and that instead its shut-off is greatly delayed, but nevertheless occurs abruptly after some stochastic lifetime with a mean of $\tau_{aberr}$. Because of the relatively small number of aberrant R* events (even at quite high flash intensities), and the fact that these events occur at stochastic locations along the outer segment, it is unfortunately necessary to simulate the activity on each disc surface, and then subsequently integrate the downstream cytoplasmic equations for the spatial case, and as a result this method is slow. On a more positive note, this approach (which we term method 3) provides an independent check on the single-compartment 'bulk' approach used in method 2.

We first needed to define the plateau level of an individual aberrant event relative to the peak of the mean single-photon event, so as to correspond approximately to the ratio reported in the literature. To accomplish this, we introduced a parameter $a_{aberr}$ that specifies the fractional R* activity in the aberrant state, with $0 < a_{aberr} \leq 1$. For the simulations presented in figures 9 and 10 we chose we set $a_{aberr} = 0.25$; see §3.5. For the other two parameters, we set $p_{aberr} = 0.002$ and $\tau_{aberr} = 4$ s for mammalian rods.

Our procedure in method 3 was to conduct repeated trials at each desired flash intensity (i.e. $\Phi$, in R* per outer segment), assigning a stochastic number $Q_{norm}$ of normal R* events, plus a stochastic number $Q_{aberr}$ (often zero) of aberrant events, to each disc surface. For those trials in which no aberrant event occurred, this procedure was essentially the same as for method 2; but for those trials in which one or more aberrant event(s) occurred, with stochastic lifetime, it was more complicated but used similar methodology. For each trial we simulated the time-course of the disc-based reactions on every disc surface. As for the other methods, the code is available online (see Data accessibility). A complete run, for 10 trials at each of 30 intensities, took 3–4 h.

## 5.11. Summary of approaches for estimating disc-based reaction kinetics

Here we briefly summarize the relative merits of the computational approaches that are applicable at different intensities.

(a) For the lowest intensities, and indeed for all intensities up to approximately 100 R*, it is appropriate to use the single-photon $E**(t)$ response, that one has calculated in advance for a large number of trials. However, it currently takes a long time (approx. 1 day on a laptop computer) to calculate a set of 2000 such single-photon responses, because it is necessary to use the full 2D diffusional approach of method 1. Thereafter, one can rapidly evaluate the downstream reactions, for any intensity up to approximately 100 R*, by integrating the set of partial differential equations for diffusion in the outer segment, and repeating this for an appropriate number of trials. In this intensity regime, there will be little error in the mean response if one simply ignores aberrant R* shut-off events.

(b) For an intermediate range of intensities, between approximately 100 R* and approximately 6000 R*, we do not currently have a fast approach, because the assumptions underlying method 2 are not applicable.

(c) For saturating flash intensities from approximately 6000 R* upwards, the analysis in §3.3 showed that method 2 (which assumes spatial homogeneity) provides predictions that are just as accurate as those of the full method 1, but that are around three orders of magnitude faster. In this approach, the depletion of transducin at very high intensities is accounted for. But, on the other hand, the late slow tail phase (that we suggest is due to aberrant R* shut-off events) is not accounted for.

(d) If one needs to describe the late slow tail phase, then it is necessary to adopt method 3, and in this case one can calculate the set of traces shown in figure 12 in a reasonable time (approx. 1 h on the laptop computer used here).

(e) For the future, we anticipate that the approach outlined in §5.7 will provide a much faster method for investigating the effects of altered parameter values on bright flash responses, at least for the regime from approximately 6000 R* upwards. Nevertheless, we expect that final calculations will need to be repeated using method 3.

(f) Finally, if one is interested in the onset phase of the response at any intensity, then the only realistic approach is to use method 1.

## 5.12. Cytoplasmic (downstream) reactions

Using the simulated PDE6 activity, we integrated the differential equations for a standard description of the downstream reactions, as set out explicitly in [11]. In the case of method 2 (the spatially homogeneous approximation method, described in §5.5), we ignored longitudinal diffusion along the outer segment, and simply considered the outer segment as a single lumped compartment. Furthermore, we solved the differential equations for the driving function given by the mean $E^{**}(t)$ activity, rather than repeatedly solving for the individual simulations and subsequently averaging. This single-compartment approach will provide only an approximate solution for just-saturating intensities (e.g. approx. 200 R* per flash), because the stochastic distribution of isomerizations along the outer segment is ignored. However, in the Results section, we found that the responses predicted in this way differ only slightly from those predicted by the alternative approach (described below) using method 3, which takes full account of the longitudinal distribution of events.

In the case of method 3 (§5.10), where aberrant R* shut-off events could occur with low probability, we took account of longitudinal diffusion in the outer segment cytoplasm. Using the simulated $E^{**}(t)$ activities in the longitudinal array of $n_x = 100$ compartments, we integrated the set of partial differential equations set out in [11], to evaluate the spatial profiles of cGMP and $Ca^{2+}$, and thereby obtain the rod's electrical response to a single trial at a single flash intensity. This procedure was repeated a number of times (typically 10 trials) to obtain an averaged flash response, and then the entire process was repeated at each intensity of interest.

The response waveforms obtained for these multiple approaches (disc-based methods 1, 2 or 3, and downstream integration for a single compartment or with longitudinal diffusion) are compared in the Results section. The close similarity of the traces that were obtained under comparable conditions, but using very different approaches, gives us confidence that there are unlikely to be serious errors in our calculations.

### 5.12.1. Cytoplasmic buffering power for cGMP

The buffering power of the cytoplasm for cyclic GMP was assumed to be $B_{cG} \approx 2$ in the original Lamb & Pugh analysis [2], but in other recent analysis such buffering has been assumed to be negligible, with $B_{cG} = 1$ [6]. Here we derive a lower limit for the buffering power, based on the known binding of cyclic GMP to the ion channels. The density of CNGC ion channels in the plasma membrane has been reported to be 200–600 channels $\mu m^{-2}$ [51,52]. We will adopt the mid value of 400 $\mu m^{-2}$, and take the plasma membrane to be a cylinder of $d = 1.3\ \mu m$ and $L = 22\ \mu m$ with an area of 90 $\mu m^2$, thereby giving a total number of 36 000 ion channels in the outer segment. Expressing this number in moles, and referencing it to the cytoplasmic volume of 0.0146 pL (table 2), we obtain an effective cytoplasmic concentration of CNGC channels of $C_{CNGC} = 4\ \mu M$. If there are $N$ sites on each CNGC that independently bind cyclic GMP with dissociation constant $K_{cG}$, then the buffering power contributed by these channels will be

$$B_{cG} = 1 + \frac{N\,C_{CNGC}}{K_{cG}}, \tag{5.22}$$

provided that the free concentration of cyclic GMP is much less than $K_{cG}$ (which it certainly will be). Although the CNGC has four binding sites for cyclic GMP, the Hill coefficient of channel opening is 3, and so we will take $N = 3$ in equation (5.22). Substituting $C_{CNGC} = 4\ \mu M$ and $K_{cG} = 20\ \mu M$ (table 2), we obtain $B_{cG} = 1.6$, as used in §4.3 in evaluating equation (4.1).

Data accessibility. The packages of computer code (WalkMat and Walk2) used to run the simulations and analyse the results in support of the findings in this article, together with sample simulated data, are included in the electronic supplementary material. Also included therein is figure S1, which collects previous experimental recordings of bright-flash responses from mammalian rods.

Authors' contributions. T.D.L. and T.W.K. conceived the study. T.D.L. wrote the computer code, ran the simulations, prepared the figures and drafted the paper. T.W.K. contributed to interpreting the data and revising the paper. Both authors read and approved the final paper.

Competing interests. We declare we have no competing interests.

Funding. This work was supported by award no. R01EY023603 from the US National Eye Institute.

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
