## [Reviewer comments · Open Biology]

Review History

RSOB-19-0241.R0 (Original submission)

Review form: Reviewer 1

Recommendation

Accept with minor revision (please list in comments)

Do you have any ethical concerns with this paper?

No

Comments to the Author

The manuscript by Lamb and Kraft presents an updated mathematical model of the mammalian rod phototransduction cascade during bright flash stimulation based on the premise of dimeric activation of rod phosphodiesterase (PDE6). The authors start by identifying a number of shortcomings of existing rod phototransduction models, most notably not taking into account the dimeric nature of the activation of PDE6 by transducin. Other shortcomings include the equivalence of recovery time constant (tail of dim flash response shutoff) and dominant time constant (measured from a series of saturating responses), inability to predict accurately the flash

intensity at which the dominant time constant changes from a lower to a higher value, and the lack of models predicting accurately the late component of bright flash responses. The authors then systematically address these issues using 3 methods: Method 1, based on stochastic simulation of 2-D diffusional interactions; a simplified Method 2 using an approximation of spatial homogeneity of disc reactants (validated for the duration of bright flash responses past the first ~100 ms; and Method 3 which incorporates the stochastic occurrence of aberrant/persistent R^* events to model the tail of bright flash responses.

The authors begin by validating the computationally faster Method 2 by comparing it with selected model traces of Model 1. Then they incorporate Model 3 and show that it produces model traces with a tail phase shutoff consistent with results from electrophysiological recordings. Finally, they use the model traces from their Models 2 and 3 to generate a plot of time in saturation vs flash intensity (Pepperberg plot) and show an excellent fit with experimental data from the literature.

Overall, this is an excellent paper, written in an accessible and insightful manner, with clearly stated conclusions derived from the method and, importantly, future experiments inspired by the results from the models. The work provides insight into several important issues in mammalian rod phototransduction, including the role of PDE dimeric activation, the rate of activation of transducin, the super linearity of the Pepperberg plot and the difference between recovery and dominant time constants, and the molecular mechanism responsible for the slow tail phase shutoff of rod responses. Its one shortcoming is that the authors do not attempt to compare the bright flash model traces from their Model 3 (Figs. 9 and 10) with actual recordings from the literature. If this is not feasible, the authors should state so and comment on the issue. The authors should also consider commenting on the implications of their model for known cases of aberrant phototransduction (i.e. rhodopsin, transducin, and PDE mutants) and perhaps even explore the issue themselves in the future.

Review form: Reviewer 2

Recommendation

Accept with minor revision (please list in comments)

Do you have any ethical concerns with this paper?

No

Comments to the Author

See attached file.

Decision letter (RSOB-19-0241.R0)

04-Nov-2019

Dear Professor Lamb,

We are pleased to inform you that your manuscript RSOB-19-0241 entitled "A quantitative account of mammalian rod phototransduction with PDE6 dimeric activation: Responses to bright flashes" has been accepted by the Editor for publication in Open Biology. The reviewer(s) have recommended publication, but also suggest some minor revisions to your manuscript. Therefore, we invite you to respond to the reviewer(s)' comments and revise your manuscript.

Please submit the revised version of your manuscript within 7 days. If you do not think you will be able to meet this date please let us know immediately and we can extend this deadline for you.

- 1) A text file of the manuscript (doc, txt, rtf or tex), including the references, tables (including captions) and figure captions. Please remove any tracked changes from the text before submission. PDF files are not an accepted format for the "Main Document".
- 2) A separate electronic file of each figure (tiff, EPS or print-quality PDF preferred). The format should be produced directly from original creation package, or original software format. Please note that PowerPoint files are not accepted.
- 3) Electronic supplementary material: this should be contained in a separate file from the main text and meet our ESM criteria (see <http://royalsocietypublishing.org/instructions-authors#question5>). All supplementary materials accompanying an accepted article will be treated as in their final form. They will be published alongside the paper on the journal website and posted on the online figshare repository. Files on figshare will be made available approximately one week before the accompanying article so that the supplementary material can be attributed a unique DOI.

Online supplementary material will also carry the title and description provided during submission, so please ensure these are accurate and informative. Note that the Royal Society will not edit or typeset supplementary material and it will be hosted as provided. Please ensure that the supplementary material includes the paper details (authors, title, journal name, article DOI). Your article DOI will be 10.1098/rsob.2016[last 4 digits of e.g. 10.1098/rsob.20160049].

- 4) A media summary: a short non-technical summary (up to 100 words) of the key findings/importance of your manuscript. Please try to write in simple English, avoid jargon, explain the importance of the topic, outline the main implications and describe why this topic is newsworthy.

Images

Data-Sharing

It is a condition of publication that data supporting your paper are made available. Data should be made available either in the electronic supplementary material or through an appropriate repository. Details of how to access data should be included in your paper. Please see <http://royalsocietypublishing.org/site/authors/policy.xhtml#question6> for more details.

Data accessibility section

Sincerely,

The Open Biology Team

<mailto:openbiology@royalsociety.org>

BM Comments to authors:

Dear Dr. Lamb,

I am pleased to inform you that your manuscript has now been evaluated by two expert reviewers that praised your work. Only some smaller details need to be address or changes before your paper is ready for publication. I have enclosed the detailed comments of the two reviewers. The point raised reviewer 1 on comparing your traces of model 3 with published traced is reasonable and also a brief discussion on the implications of your model on the evaluation of abnormal phototransduction. Reviewer 2 has many specific comments I will leave at your discretion to address in your manuscript.

Congratulations and best regards

Stephan Neuhauss

Reviewer(s)' Comments to Author:

Referee: 1

Comments to the Author(s)

The manuscript by Lamb and Kraft presents an updated mathematical model of the mammalian rod phototransduction cascade during bright flash stimulation based on the premise of dimeric activation of rod phosphodiesterase (PDE6). The authors start by identifying a number of shortcomings of existing rod phototransduction models, most notably not taking into account the dimeric nature of the activation of PDE6 by transducin. Other shortcomings include the equivalence of recovery time constant (tail of dim flash response shutoff) and dominant time constant (measured from a series of saturating responses), inability to predict accurately the flash

intensity at which the dominant time constant changes from a lower to a higher value, and the lack of models predicting accurately the late component of bright flash responses. The authors then systematically address these issues using 3 methods: Method 1, based on stochastic simulation of 2-D diffusional interactions; a simplified Method 2 using an approximation of spatial homogeneity of disc reactants (validated for the duration of bright flash responses past the first ~100 ms; and Method 3 which incorporates the stochastic occurrence of aberrant/persistent R* events to model the tail of bright flash responses.

The authors begin by validating the computationally faster Method 2 by comparing it with selected model traces of Model 1. Then they incorporate Model 3 and show that it produces model traces with a tail phase shutoff consistent with results from electrophysiological recordings. Finally, they use the model traces from their Models 2 and 3 to generate a plot of time in saturation vs flash intensity (Pepperberg plot) and show an excellent fit with experimental data from the literature.

Overall, this is an excellent paper, written in an accessible and insightful manner, with clearly stated conclusions derived from the method and, importantly, future experiments inspired by the results from the models. The work provides insight into several important issues in mammalian rod phototransduction, including the role of PDE dimeric activation, the rate of activation of transducin, the super linearity of the Pepperberg plot and the difference between recovery and dominant time constants, and the molecular mechanism responsible for the slow tail phase shutoff of rod responses. Its one shortcoming is that the authors do not attempt to compare the bright flash model traces from their Model 3 (Figs. 9 and 10) with actual recordings from the literature. If this is not feasible, the authors should state so and comment on the issue. The authors should also consider commenting on the implications of their model for known cases of aberrant phototransduction (i.e. rhodopsin, transducin, and PDE mutants) and perhaps even explore the issue themselves in the future.

Referee: 2

Comments to the Author(s)
See attached file.

Author's Response to Decision Letter for (RSOB-19-0241.R0)

See Appendix A.

Decision letter (RSOB-19-0241.R1)

21-Nov-2019

Dear Professor Lamb,

We are pleased to inform you that your manuscript entitled "A quantitative account of mammalian rod phototransduction with PDE6 dimeric activation: Responses to bright flashes" has been accepted by the Editor for publication in Open Biology.

Article processing charge

Please note that the article processing charge is immediately payable. A separate email will be sent out shortly to confirm the charge due. The preferred payment method is by credit card; however, other payment options are available.

Sincerely,

The Open Biology Team
mailto: openbiology@royalsociety.org

RSOB-19-0241 - Response to Reviewers' comments

We thank both reviewers for their highly positive reports. In the following text we have extracted their criticisms and suggestions, and we have responded to all of those.

Reviewer 1

Thank you very much for such a positive report. Below are the two criticisms/suggestions and our responses:

Its one shortcoming is that the authors do not attempt to compare the bright flash model traces from their Model 3 (Figs. 9 and 10) with actual recordings from the literature.

Excellent suggestion. We have now added a new Figure 12 making this comparison.

The authors should also consider commenting on the implications of their model for known cases of aberrant phototransduction (i.e. rhodopsin, transducin, and PDE mutants) and perhaps even explore the issue themselves in the future.

Another great idea. We have added a new short Section 4.5 in the Discussion.

Reviewer 2

Thank you very much for such a positive report. Below are the various criticisms/suggestions and our responses:

The following somewhat negative comments do not detract substantially from the paper's merits. The paper is a bit long. Perhaps it could be streamlined a bit.

We feel that we cannot shorten the paper without detracting from the scientific content.

There are no comparisons with laboratory electrophysiological recordings.

This was also Reviewer 1's main point. We have added a new Figure 12 making such comparison for one family from the literature. The only published recordings (that we have been able to find) at these intensities are presented in the Supplementary Figures. However, we do not have the raw data, and nor do we have accurate values for the individual flash intensities in any of these families. For the future, our aim is to make our own experimental recordings from mammalian rods exposed to very bright flashes, and to test the model against such data.

All, or nearly all, the simulated responses are supersaturating responses. The problem is that the bright flash responses presented in the dark do not fully characterize the rod response properties. I can understand that this paper is specifically about the responses to bright flashes, but it leaves open the question of whether the model can account for sub-saturating responses and families of responses to flashes superimposed on background light levels spanning the full physiological range. It seems that papers of Fain and coworkers, who did such measurements and modeling, are germane to the present paper. The omission of these references is surprising.

Yes, this paper is concerned primarily with responses to bright flashes presented in darkness. We agree that it will be necessary in the future to investigate the effect of background

illumination, but such an investigation of ‘light adaptation of bright flash responses’ is beyond the scope of this study. Accordingly, the appropriate place for presenting, modelling, and discussing such phenomena will be a future paper.

(1) Ramping: Is it possible to understand this in terms of the model equations taking diffusion into account? Why a ramp; why 2 exponential filter stages? Can the empirical-equation approach be extended, both to higher numbers of absorbed photons Q and out to later time points? The success of the empirical equation (5.1) calls into question the need to do full stochastic simulations on each disc surface, at least under certain conditions.

We have added a short paragraph in Section 3.2, to try to explain these issues.

Incidentally, the shorthand phrase "convolved with a time constant" is used in a number of places, and it would be better to say, "convolved with an exponential function of time with time constant" Agreed, and done.

(2) The four-fold increase in the authors' estimate of the rate of transducin activation per R^* over the widely accepted value is great to see. Thank you.

(3) Perhaps the single-photon response material could be put in the supplemental information/methods section.

We looked at this possibility, but we feel that it detracts from the reader's ability to see that the same set of parameters can be used to explain the responses in both regimes, dim and bright.

(4) Section 3.3.1 is an example of a section in which the reader might be uncertain about the meaning of Method 2

While we appreciate the issues raised by the Reviewer, we can't see that it applies to Section 3.3.1, and we wonder whether this might be meant to be Section 5.3.1. In any case, we have attempted to deal with the issue through our changes in response to point (9) below.

(5) I'm wondering if there is a conceptual mistake on the application of probability theory to the case of multiple photoisomerizations Q on a disc surface in section 5.4.2. ... The sum of lifetimes does not seem to me to be a random variable of interest in this situation.

The reason for calculating the sum of R^* lifetimes is because, in the binary stochastic shut-off model, each R^* has unit activity prior to its shut-off, so that an estimate of the ‘total R^* activity’ can be obtained simply by summing the Q lifetimes. This approximation is reasonable because the mean R^* lifetime (68 ms) is short in comparison with the response times being examined for bright flashes. A sentence has been added to explain this.

(6) Figure 7 on the super-linearity of $E^{**}(t)$ responses at 0.7 s differs from the linear behavior in Figure 3 for much earlier times. I don't understand this super-linear behavior. I'm wondering if the authors can say more about it by way of giving an explanation or an intuitive understanding.

This is a good point, and we have added a paragraph at the end of Section 3.3.3, to contrast the linearity found at early times with the super-linearity found at late times, and to provide an intuitive explanation.

(7) The section on time spent in saturation is terrific. Thank you.

(8) Section 5.9 on the depletion of G-protein is stunning. Thank you.

I suggest reversing the order of Equations (5.21) and (5.20). Also, I'm wondering if the authors have a motivation for their Equation (5.21). The authors seem to imply that the average photocurrent response can be computed from the average G^* time course for each disc surface. It is not obvious that this would be the case.

As requested, we have reversed the order of these equations, and reworded the text. We use the average time-course $G(t)$ of available transducin in this equation simply for the purpose of determining depletion of the rate at which G^* can be activated. The Reviewer is probably correct that it might be better to determine the depletion on individual discs, but that would be more complicated. And, as this correction for depletion only makes any difference at the very highest intensities, we think that the added complexity would not be worth it.

(9) Finally, perhaps it would be useful to have some sort of grand reckoning on what the fastest but accurate methods that can be use in various time and photoisomerizations regimes. It is easy to lose track in this paper.

Thank you for this suggestion. In order to accomplish this, we have added a new Section 5.11 to summarise the methods that are appropriate at different intensities.

It is a flaw that the model was not used to fit any real families of supersaturating responses from individual rods. Perhaps this is asking too much.

As mentioned above, we have added a new Figure 12 to compare model and experiment for one such family of supersaturating responses. And, for the future, our aim is to make our own experimental recordings from mammalian rods exposed to very bright flashes, and to further test the model against these data.